# Characterizing the mechanism of action for mRNA therapeutics for the treatment of propionic acidemia, methylmalonic acidemia, and phenylketonuria

Rena Baek[1], Kimberly Coughlan[1], Lei Jiang[1], Min Liang [1], Lei Ci[1], Harkewal Singh[1], Hannah Zhang[1], Neeraj Kaushal[1], Ivana Liric Rajlic[1], Linh Van[1], Rain Dimen[1], Alexander Cavedon[1], Ling Yin[1], Lisa Rice[1], Andrea Frassetto[1], Lin Guey[1] ✉, Patrick Finn [1] ✉ & Paolo G. V. Martini [1] ✉

Messenger RNA (mRNA) therapeutics delivered via lipid nanoparticles hold the potential to treat metabolic diseases caused by protein deficiency, including propionic acidemia (PA), methylmalonic acidemia (MMA), and phenylketonuria (PKU). Herein we report results from multiple independent preclinical studies of mRNA-3927 (an investigational treatment for PA), mRNA-3705 (an investigational treatment for MMA), and mRNA-3210 (an investigational treatment for PKU) in murine models of each disease. All 3 mRNA therapeutics exhibited pharmacokinetic/pharmacodynamic (PK/PD) responses in their respective murine model by driving mRNA, protein, and/or protein activity responses, as well as by decreasing levels of the relevant biomarker(s) when compared to control-treated animals. These preclinical data were then used to develop translational PK/PD models, which were scaled allometrically to humans to predict starting doses for first-in-human clinical studies for each disease. The predicted first-in-human doses for mRNA-3927, mRNA-3705, and mRNA-3210 were determined to be 0.3, 0.1, and 0.4 mg/kg, respectively.

Messenger RNA (mRNA) therapeutics consist of specific mRNA construct sequences encapsulated in lipid nanoparticles (LNPs). LNPs are designed with specific ratios of ionizable lipids, cholesterol, phospholipids, and polyethylene glycol–conjugated lipids (Fig. 1a)[1]. The chemical and physical composition of the LNPs and mRNA constructs facilitate biodistribution, cellular uptake, and endosomal escape, with the goal of driving protein expression to facilitate the desired therapeutic effect (Fig. 1b)[1]. Additional features that contribute to the success of an mRNA-based therapeutic include translation initiation fidelity, functional mRNA half-life, targeting of the desired cell type, and tolerable immunogenicity[1]. Because of their ability to drive protein expression, mRNA therapies offer a promising strategy for the

treatment of diseases caused by protein deficiency[2]. Three such disorders include the following rare, inherited, metabolic diseases: propionic acidemia (PA), methylmalonic acidemia (MMA), and phenylketonuria (PKU)[3–5].

PA is caused by pathogenic variants in the propionyl-coenzyme A (CoA) carboxylase (PCC) α and/or β subunit genes (*PCCA* and *PCCB*, respectively), which encode a mitochondrial enzyme that catalyzes the conversion of propionyl-CoA to methylmalonyl-CoA in the propionate metabolism pathway[6]. Deficiencies in PCC lead to an accumulation of toxic metabolites, including the organic acids 2-methylcitrate (2-MC) and 3-hydroxypropionate (3-HP), and increases in the ratio of propionylcarnitine to acetylcarnitine (C3/C2)[7–9]. Patients with PA

[1]Moderna, Inc., 200 Technology Square, Cambridge, MA 02139, USA. ✉e-mail: Lin.Guey@modernatx.com; Patrick.Finn@modernatx.com; Paolo.Martini@modernatx.com

**Fig. 1 | LNP nucleic acid delivery of therapeutic mRNA. a** LNP structure.
**b** Approach to PA, PKU, and MMA treatment. Δ change, LNP lipid nanoparticle,
MMA methylmalonic acidemia, mRNA messenger RNA, PA propionic acidemia, PEG
polyethylene glycol, PKU phenylketonuria. [a]Adapted from An et al.[4] with permission from Elsevier. Source data are provided as a Source Data file.

frequently experience life-threatening acute metabolic decompensation events within the first few years of life, while those who survive can suffer from growth retardation, chronic renal failure, and neurologic complications[10].

MMA is primarily caused by a defect or deficiency in methylmalonyl-CoA mutase (MUT), an enzyme that functions just downstream of PCC in the propionate metabolic pathway to catalyze the reversible isomerization of L-methylmalonyl-CoA to succinyl-CoA. A deficiency in MUT blocks the propionate metabolism pathway, resulting in the accumulation of toxic metabolites, including

methylmalonic acid, 2-MC, and C3[11]. Given the defects in a shared metabolic pathway, patients with MMA experience similar outcomes as those with PA[11–14].

PKU is an autosomal recessive disorder that results in defects in phenylalanine hydroxylase (PAH), the enzyme responsible for conversion of phenylalanine (Phe) to tyrosine[5]. Defects in PAH lead to hyperphenylalaninemia, which can disrupt neurologic function, causing severe intellectual disability, epilepsy, and psychiatric problems.[5]

The current standards of care for each of these disorders center on symptom management or avoiding dietary components that

exacerbate disease. For example, recommendations for patients with PA and MMA include management of acute metabolic decompensation events and diets low in natural proteins to reduce the precursor molecules that funnel into the defective metabolic pathways[15]. While liver transplants for patients with PA and combined liver-kidney transplants for those with MMA have been shown to reduce the number of metabolic decompensations and improve biochemical markers, this practice does not eliminate the risk of disease-related complications[15,16]. For patients with PKU, a Phe-restricted diet with pegvaliase-pqpz, tyrosine, and/or tetrahydrobiopterin supplementation are recommended[5].

Rather than mitigating symptoms, other types of therapeutics, such as gene therapies, are being explored to address the underlying causes of these disorders. Canonical adeno-associated virus gene delivery has shown promise in murine models of PA, MMA, and PKU; however, such treatments can be subject to genotoxicity in the form of carcinoma formation and immunologic responses or preexisting immunity to the viral vector[17–20]. Unlike gene delivery by viral vectors, preclinical studies of LNP-mRNA therapies have demonstrated minimal toxicity, negligible inflammatory or immune responses, and sustained safety and tolerability, even with repeated exposure[4,21].

Previous reports have provided proof of concept for mRNA-based therapies in murine models of PA and MMA[4,21]. Pharmacology data from murine studies of PA were used to develop a translational semimechanistic pharmacokinetic (PK) and PK/pharmacodynamic (PD) model to quantify the relationship between the dose of mRNA-3927, an investigational PCCA/B mRNA treatment for PA, and PK/PD responses[3]. The model adequately described the plasma PK of PCCA/B mRNA in mice, rats, and monkeys. Similar modeling related to mRNA therapeutics for MMA and PKU has not been reported. Here we report results from multiple independent preclinical studies of mRNA-based treatments delivered via LNPs, including mRNA-3927, mRNA-3705 (an investigational treatment for MMA), and mRNA-3210 (an investigational treatment for PKU). These preclinical data were used to develop translational semimechanistic PK/PD models, which were scaled allometrically to humans to predict starting doses for first-in-human (FIH) clinical studies in PA, MMA, and PKU populations.

## Results
Several independent preclinical studies were conducted in murine models of PA, MMA, and PKU. Given the differences in these models and the distinct requirements for PK/PD modeling components, a conglomerate of factors, including mRNA concentrations, protein concentrations, protein activity levels, and/or the quantity of biomarkers, were assessed following treatment with LNP-formulated mRNA therapeutics in each of the specified mouse models.

### Pcca⁻/⁻ (A138T) mice treated with LNP-formulated hPCCA and hPCCB mRNA
In the PA PK study, $Pcca^{-/-}$ (A138T) mice were treated with a single intravenous (IV) bolus dose of LNP-encapsulated human PCCA (hPCCA) and human PCCB (hPCCB) mRNA (mRNA-3927). Plasma concentrations of hPCCA and hPCCB mRNA from mRNA-3927 were quantifiable up to 48 h postdose in the 1.0 and 2.0 mg/kg dose groups (Fig. 2a, b). PK parameters (including area under the concentration versus time curve from the last start of dose administration to the time after dosing [$AUC_{tlast}$], maximum observed concentration [$C_{max}$], terminal elimination half-life, clearance rate, and volume of distribution) were generally similar between hPCCA and hPCCB mRNA (Supplementary Table 1). The exposure of hPCCA and hPCCB mRNA increased with increasing dose and appeared to be independent of sex. The fold-increase in exposure was approximately proportional to the fold-increase in dose level.

In the PA PD study, $Pcca^{-/-}$ (A138T) mice treated with 2 IV bolus doses of mRNA-3927 at 0.5 mg/kg exhibited a nonsignificant increase

in mean ± standard deviation (SD) PCC activity compared to Tris-sucrose control mice ($7.77 \pm 4.27$ and $2.72 \pm 2.26$ ng/mg protein, respectively; Fig. 2c). Accordingly, PD responses, defined by reductions from baseline in plasma 2-MC, 3-HP, and the ratio of C3/C2, were observed in treated mice, while no metabolic response was observed in the controls (Fig. 2d–f and Supplementary Table 2). The duration of biochemical responses after treatment with the first IV dose of hPCCA and hPCCB mRNAs was ~3 to 4 weeks for all plasma biomarkers.

### CD1 and Mut⁻/⁻;Tg^INS-CBA-G715V mice treated with LNP-formulated hMUT mRNA
In the MMA PK study, CD1 mice were treated with a single IV bolus dose of LNP-encapsulated human MUT (hMUT) mRNA (mRNA-3705) at a 0.1, 0.5, or 1.0 mg/kg dose. Plasma concentrations of hMUT mRNA from mRNA-3705 were quantifiable up to 24 h after dosing at 0.1 mg/kg for males and throughout the 48-hour sampling period at 0.1 mg/kg for females and 0.5 and 1.0 mg/kg in both sexes (Fig. 3a). The systemic exposure of hMUT mRNA from mRNA-3705 increased with doses for $C_{max}$ and $AUC_{tlast}$ between 0.1 and 1.0 mg/kg (Supplementary Table 3). Dose-normalized plasma exposures were generally lower in females than in males at 0.1 and 0.5 mg/kg. The plasma $C_{max}$ and $AUC_{tlast}$ values determined for hMUT mRNA from mRNA-3705 showed no apparent sex difference at 1.0 mg/kg dose of mRNA-3705.

In the MMA PD study, $Mut^{-/-}$;Tg^INS-CBA-G715V hypomorphic mice were treated with 0.2 and 0.5 mg/kg doses of mRNA-3705. Mean ± SD liver hMUT protein concentrations were expressed at a significantly higher level compared to phosphate-buffered saline (PBS) controls (0.2 mg/kg mRNA-3705: $45.0 \pm 42.2$ ng/mg; 0.5 mg/kg mRNA-3705: $70.4 \pm 24.2$ ng/mg; control: no protein detected; Fig. 3b and Supplementary Table 4). Mean ± SD hepatic MUT enzyme activity was significantly increased in $Mut^{-/-}$; Tg^INS-CBA-G715V hypomorphic mice treated with 0.5 mg/kg mRNA-3705 compared to the controls (Fig. 3c). Mean ± SD MUT enzyme activity in liver samples was $5.7 \pm 2.8$ nmol/min/mg protein in the group treated with 0.2 mg/kg mRNA-3705 and $10.7 \pm 6.5$ nmol/min/mg protein in the group treated with 0.5 mg/kg mRNA-3705. MUT enzyme activity in the control mice was $2.8 \pm 0.2$ nmol/min/mg protein. While WT mice were not included in this study, subsequent studies that included unaffected heterozygous $Mut^{+/-}$ littermates demonstrated that treatment of $Mut^{-/-}$;Tg^INS-CBA-G715V with mRNA-3705 at 1.0 mg/kg resulted in approximately 50% of MUT activity compared with $Mut^{+/-}$ mice 24 h postdose.

At 24 h postdose, $Mut^{-/-}$;Tg^INS-CBA-G715V hypomorphic mice treated with 0.5 mg/kg of mRNA-3705 exhibited a significant decrease (mean ± SD: 78% ± 3.2% reduction from baseline) in plasma methylmalonic acid concentrations (Fig. 3d). No metabolic response was observed in control mice. Mean ± SD liver, kidney, and heart methylmalonic acid concentrations were also significantly lower in both groups of mice treated with hMUT mRNA 24 h postdose compared to the control group.

### PAH^enu2 mice treated with LNP-formulated hPAH mRNA
Following a single IV dose of an LNP-formulated human PAH (hPAH) mRNA (mRNA-3210) at 0.25, 0.5, and 1.0 mg/kg in the PKU PK/PD study, mean ± SD serum and liver concentrations of mRNA from mRNA-3210 were quantifiable for up to 168 hours (Fig. 4a, b). In serum, the $C_{max}$ of mRNA from mRNA-3210 was observed at the first collection time point (0.25 h postdose) in female mice and at 0.25 to 1 h postdose in male mice. At the time of the observed $C_{max}$ ($T_{max}$) in female mice, serum concentrations of mRNA from mRNA-3210 at 0.25, 0.5, and 1.0 mg/kg doses were 434, 959, and 1850 ng/mL all at 0.25 h postdose, respectively. In male mice, serum concentrations at $T_{max}$ were 453, 876, and 1,640 ng/mL at 0.25, 1, and 1 h postdose, respectively (Supplementary Table 5). The $C_{max}$ of mRNA from mRNA-3210 in liver samples occurred at 2 h postdose at all dose levels, independent of sex

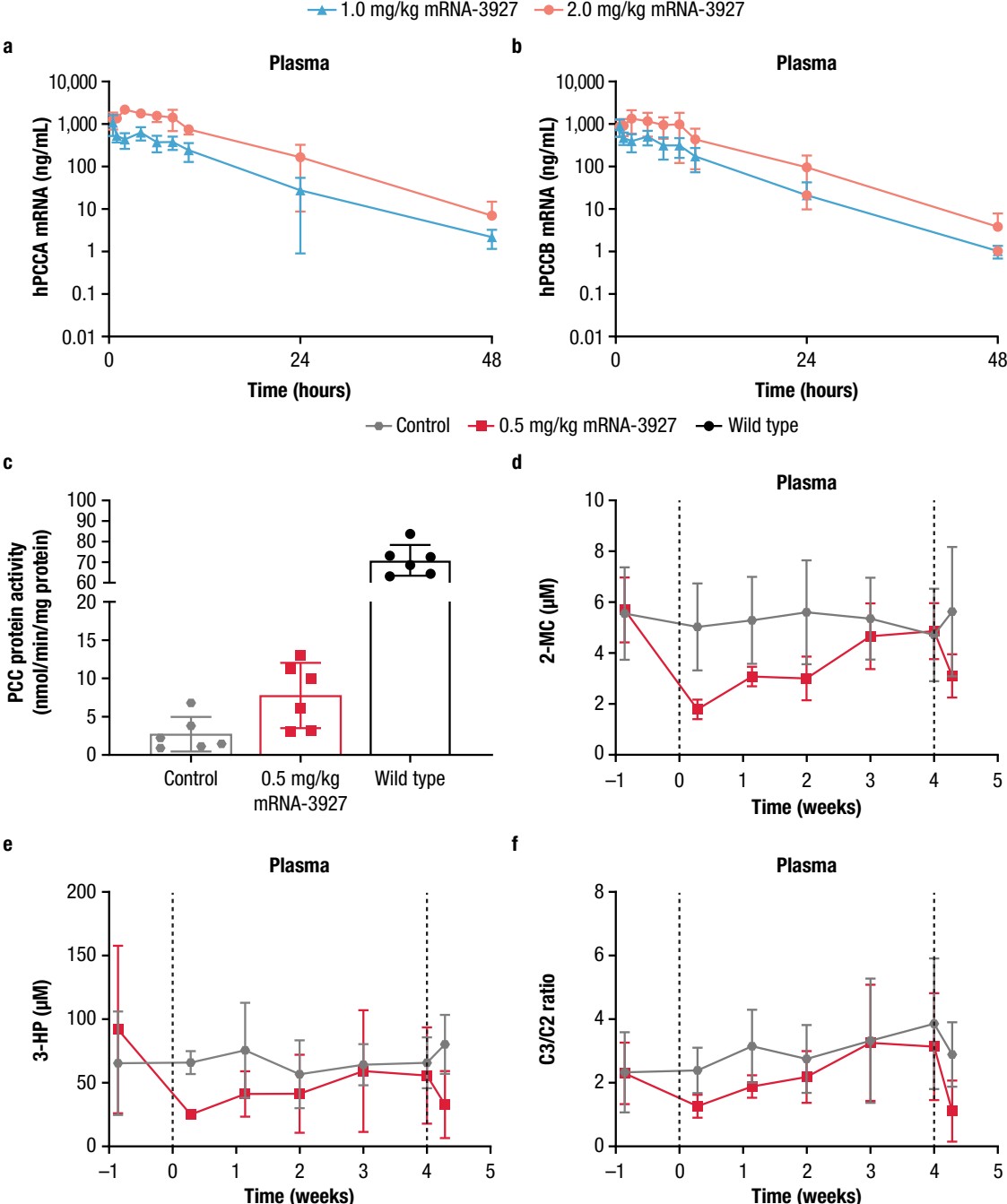

**Fig. 2 | Mean (±SD) hPCCA and hPCCB mRNA concentrations, PCC enzyme activity, and metabolite concentrations in Pcca⁻/⁻ (A138T) mice treated with mRNA-3927. a, b** *Pcca⁻/⁻* (A138T) mice received a single bolus IV dose of mRNA-3927 (1.0 or 2.0 mg/kg). Blood samples were collected at multiple time points up to 48 h postdose. Plasma concentrations of hPCCA (*n* = 4–7 mice/dose level/time-point) and hPCCB mRNA (*n* = 4–7 mice/dose level/timepoint), respectively were quantified using branched DNA analysis. **c** *Pcca⁻/⁻* (A138T) mice received 2 IV bolus doses of Tris-sucrose (control) or 0.5 mg/kg mRNA-3927 on Days 0 and 28 (*n* = 6 mice/group). PCC activity was measured using a radiometric activity assay wherein the mitochondrial fractions of liver homogenates were incubated with a PCC substrate and the enzymatic product was quantified by scintillation. **d–f** Blood was collected from mice prior to treatment and on Days 2, 8, 14, 22, and 28 after the first IV dose, and 2 days after the second IV dose. Plasma concentrations of 2-MC (*n* = 6 mice/group, where only *n* = 5 samples were available for analysis in the 0.5 mg/kg

mRNA-3927 group at Day 8), 3-HP (*n* = 6 mice/group, where only *n* = 2 and 5 samples were available for analysis in the Wild type group at Days 2 and 22, respectively; where only *n* = 5 and 2 samples were available for analysis in the 0.5 mg/kg mRNA-3927 group prior to treatment and Day 8, respectively; where only *n* = 5 samples were available for analysis in the Control group at Day 2), and C3/C2 (*n* = 6 mice/group, where only *n* = 5 samples were available for analysis in the 0.5 mg/kg mRNA-3927 group at Days 8) were quantified by LC-MS/MS. Dotted lines represent dose administrations. 2-MC 2-methylcitrate, 3-HP 3-hydroxypropionate, C3/C2 ratio of propionylcarnitine to acetylcarnitine, hPCCA human propionyl-coenzyme A carboxylase α subunit, hPCCB human propionyl- coenzyme A carboxylase β subunit, IV intravenous, LC-MS/MS liquid chromatography-tandem mass spectrometry, mRNA messenger RNA, PCC propionyl- coenzyme A carboxylase, SD standard deviation. Source data are provided as a Source Data file.

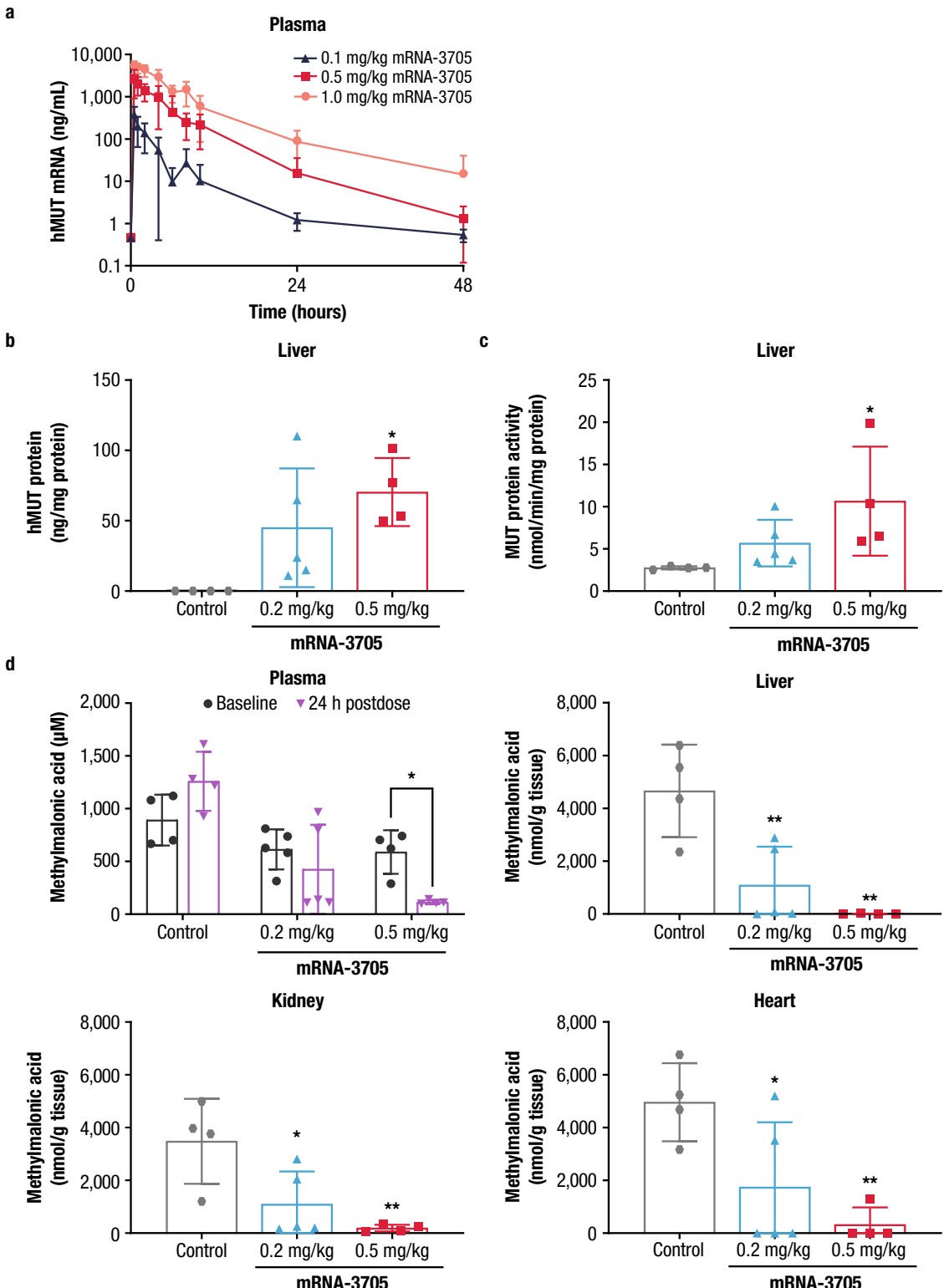

(Supplementary Table 6). Mean ± SD concentrations of hPAH protein were quantifiable in the liver samples of mice receiving 0.25 mg/kg mRNA-3210 for up to 96 h postdose, and for up to 168 h postdose in those receiving 0.5 and 1.0 mg/kg mRNA-3210 (Fig. 4c). The maximum effect of liver hPAH protein occurred at 24 h postdose in mice treated with 0.25 and 0.5 mg/kg mRNA-3210 and at 8 h postdose in mice treated with 1.0 mg/kg mRNA-3210. Exposures for hPAH protein increased with increasing doses (Fig. 4c). At $TE_{max}$, liver protein concentrations at 0.25, 0.5, and 1.0 mg/kg doses were 6950, 16,700, and

35,600 ng/g, respectively (Supplementary Table 7). Phe blood concentrations (mean ± SD) were sustained and similar in control $PAH^{enu2}$ mice and those receiving a single dose of 0.25 mg/kg mRNA-3210 throughout the sampling period (Fig. 4d). After administration of a single dose of mRNA-3210 at 0.5 and 1.0 mg/kg, dose-dependent reductions of endogenous Phe concentrations were observed. The median time to maximum effect (corresponding to minimum Phe blood concentration) in mice treated with 0.5 and 1.0 mg/kg of mRNA-3210 was observed by 24 and 48 h postdose, respectively, with a

**Fig. 3 | Mean (±SD) concentrations of hMUT mRNA in CD1 mice and hMUT protein, enzyme activity, and methylmalonic acid levels in *Mut*$^{-/-}$;Tg$^{INS-CBA-G715V}$ hypomorphic mice treated with a single dose of mRNA-3705. a** CD1 mice received a single IV injection of mRNA-3705 (0.1, 0.5, or 1.0 mg/kg). Blood samples were collected on Day 1 at multiple time points up to 48 h postdose for hMUT mRNA quantitation via a bDNA method analysis ($n = 6$ mice/timepoint for all groups at 0.5, 1, 6, 8, 10, 24, and 48 hours postdose; and $n = 12$ mice/timepoint for all groups at 2 and 4 h postdose). **b** *Mut*$^{-/-}$;Tg$^{INS-CBA-G715V}$ hypomorphic mice received a single IV injection of either PBS (control) or mRNA-3705 (0.2 or 0.5 mg/kg). mRNA-encoded hMUT protein concentrations were quantified in the liver of mice receiving the indicated treatment 24 h postdose using a human-specific LC-MS/MS method ($n = 4$, 5, and 4 biologically independent mice per Control and 0.2 mg/kg and 0.5 mg/kg mRNA-3705, respectively). **c** A MUT activity assay that did not differentiate between human versus murine enzyme activity was used to assess total

MUT enzyme activity ($n = 4$, 5, and 4 biologically independent mice per Control and 0.2 mg/kg and 0.5 mg/kg mRNA-3705, respectively). **d** Plasma methylmalonic acid levels were measured at baseline (Day -7) and 24 h postdose in mice receiving the indicated treatment ($n = 4$, 5, and 4 biologically independent mice per Control and 0.2 mg/kg and 0.5 mg/kg mRNA-3705, respectively). Liver, kidney, and heart methylmalonic acid levels were measured at 24 h postdose ($n = 4$, 5, and 4 biologically independent mice per Control and 0.2 mg/kg and 0.5 mg/kg mRNA-3705, respectively). *$P < 0.05$, **$P < 0.01$ compared to *Mut*$^{-/-}$;Tg$^{INS-CBA-G715V}$ PBS group by 1-way ANOVA, followed by Dunnett's multiple comparisons test. ANOVA analysis of variance, bDNA branched DNA, hMUT human methylmalonyl-coenzyme A mutase, IV intravenous, LC-MS/MS liquid chromatography-tandem mass spectrometry, mRNA messenger RNA, PBS phosphate-buffered saline, SD standard deviation. Source data are provided as a Source Data file.

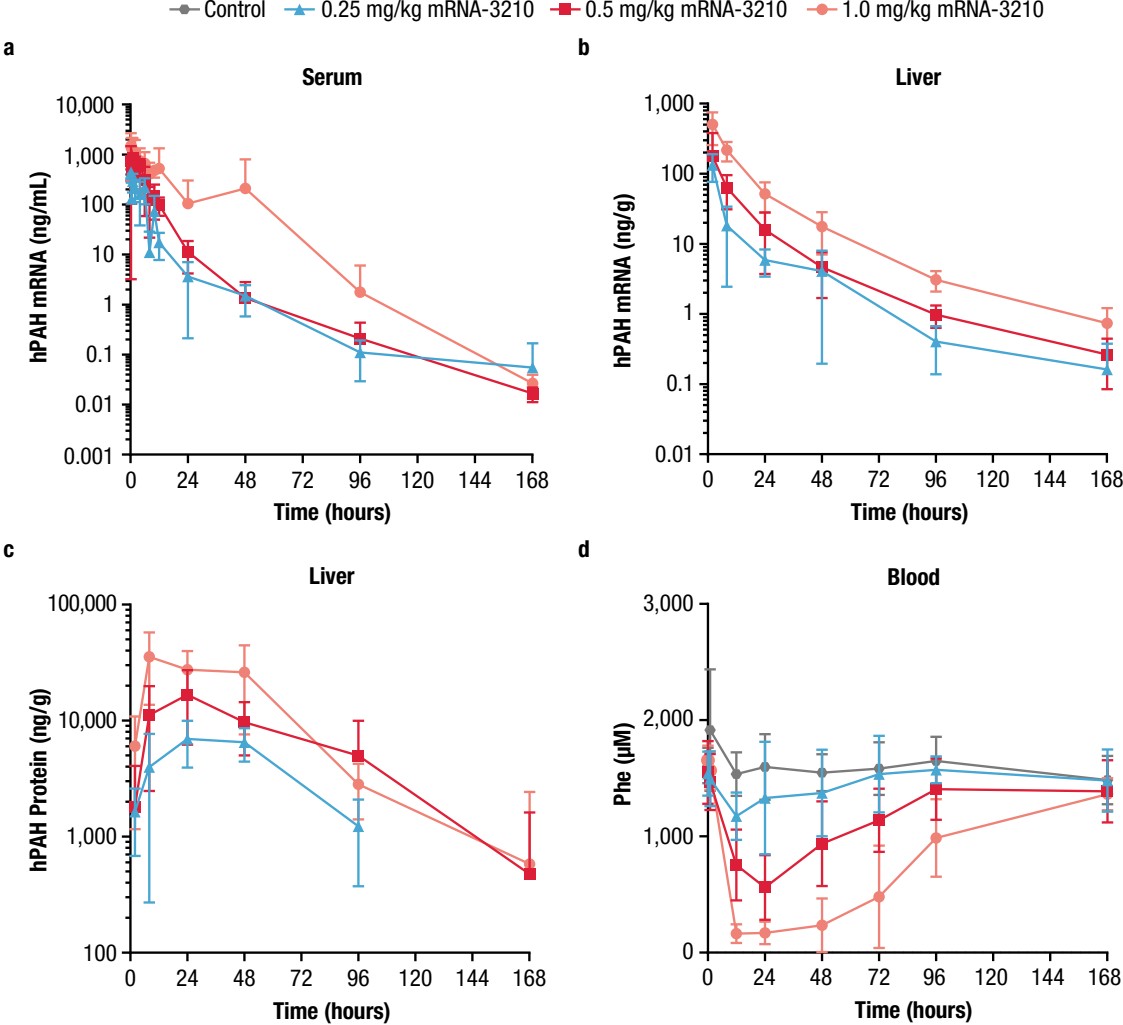

**Fig. 4 | Mean (±SD) serum and liver concentrations of hPAH mRNA, hPAH protein, and blood Phe levels in PAHenu2 mice treated with a single IV dose of mRNA-3210. a–d** PAH$^{enu2}$ mice ($n = 6$–16/treatment group) were treated with a single IV dose of mRNA-3210, and serum ($n = 9$ mice for 0.25 mg/kg mRNA-3210 at 0.25 h postdose and $n = 8$ mice for all other timepoints; $n = 9$ mice for 0.5 mg/kg mRNA-3210 at 0.5 h postdose and $n = 8$ mice for all other timepoints; $n = 9$ mice for 1.0 mg/kg mRNA-3210 at 0.25 h postdose and $n = 8$ mice for all other timepoints) and liver hPAH mRNA ($n = 9$ mice for 0.25 mg/kg mRNA-3210 at 2 h postdose, $n = 8$ mice at 8, 24, 48, 96 h postdose, $n = 8$ mice at 8 h postdose, and $n = 16$ mice at 168 h postdose; $n = 8$ mice for 0.5 mg/kg mRNA-3210 at 2, 24, 48, 96 h postdose, $n = 9$ mice at 8 h postdose, and $n = 16$ mice at 168 h postdose; $n = 9$ mice for 1.0 mg/kg mRNA-3210 at 2 h postdose, $n = 8$ mice at 8, 24, 48, 96 h postdose, $n = 8$ mice at 8 h postdose, and $n = 16$ mice at 168 h postdose) or hPAH protein concentrations

($n = 10$ mice for 0.25 mg/kg mRNA-3210 at 2 h postdose, $n = 8$ mice at 8, 24, 48, 96 h postdose, $n = 7$ mice at 96 h postdose, and $n = 16$ mice at 168 h postdose; $n = 8$ mice for 0.5 mg/kg mRNA-3210 at 2, 24, 48, 96 h postdose, $n = 9$ mice at 8 h postdose, and $n = 16$ mice at 168 h postdose; $n = 9$ mice for 1.0 mg/kg mRNA-3210 at 2 h postdose, $n = 8$ mice at 8, 24, 48, 96 h postdose, $n = 16$ mice at 168 h postdose), were assessed at baseline and 6, 24, 48, 72, 96, and 168 h postdose. **d** PAH$^{enu2}$ mice ($n = 8$/ treatment group) were treated with a single IV dose of mRNA-3210 at 0.25, 0.5, or 1.0 mg/kg or saline (control). Blood Phe levels were assessed at baseline and 6, 24, 48, 72, 96, and 168 h postdose. enu2 N-ethyl-N-nitrosourea, hPAH human phenylalanine hydroxylase, IV intravenous, mRNA messenger RNA, PAH phenylalanine hydroxylase, Phe phenylalanine, PKU phenylketonuria, SD standard deviation. Source data are provided as a Source Data file.

reduction in the area under the effective curve from 0 to 168 h of 26.3% and 51.5%, respectively. At maximum effect, mean ± SD Phe blood concentrations at 0.25, 0.5, and 1.0 mg/kg doses were 1110 ± 209, 561 ± 279, and 138 ± 78 µmol/L, respectively. Wild-type (WT) mice were not included in this study; however, subsequent studies including vehicle-treated WT mice showed that Phe levels were around 100 µM for the duration of the study. At all dose levels, Phe concentrations (mean ± SD) returned to baseline levels by 168 h postdose (Fig. 4d).

### Translational PK/PD modeling

The translational strategy in selecting starting clinical doses for mRNA-3927, mRNA-3705, and mRNA-3210 consisted of FIH dose projections, where PK/PD models were developed for each mRNA therapeutic. Accordingly, the preclinical data, including mRNA, protein, and biomarker time-concentration profiles described above were used to independently develop preclinical PK/PD models. Each preclinical PK/PD model was extrapolated to humans based on allometric scaling of model parameters. For PA and MMA, mouse, rat, and monkey data were available to utilize an interspecies scaling approach to project FIH doses, whereas for PKU, the relevant disease PAH$^{enu2}$ mouse model was only considered for FIH dose projections. Model-based simulations were performed in support of FIH dose projections (Fig. 5). All 3 programs used the same LNP (Table 1). The predicted FIH doses for the PA, MMA, and PKU programs were 0.3, 0.1, and 0.4 mg/kg, respectively. These values were well below the no-observed-adverse-effect-limit (PA: 3 mg/kg; MMA: 5 mg/kg; and PKU: 3 mg/kg), with safety margins of 10, 50, and 7.5, respectively.

## Discussion

The standard of care for PA, MMA, and PKU have traditionally focused on symptom management to avoid metabolic crisis and neurologic impairment. mRNA therapies offer a revolutionary approach to directly address the underlying defects responsible for these disorders by providing the instructions for the production of functional proteins that are deficient or nonfunctional due to genetic mutation. These nonclinical studies evaluating the PK of mRNA-3927, mRNA-3705, and mRNA-3210 demonstrated that, following IV administration, mRNAs derived from the respective therapeutic could be detected in murine models of PA, MMA, and PKU, respectively. Furthermore, a PD analysis demonstrated the ability of each mRNA therapeutic to reduce the concentration of the pertinent primary biomarkers of disease. In *Pcca*$^{-/-}$ (A138T) mice treated with mRNA-3927, plasma 2-MC, 3-HP, and C3/C2 ratios were reduced for up to 3 to 4 weeks following the first IV dose. Single IV administration of mRNA-3705 in *Mut*$^{-/-}$;Tg$^{INS-CBA-G715V}$ hypomorphic mice resulted in a significant decrease of methylmalonic acid in tissues that are affected in MMA and associated with common clinical findings in patients with MMA, including the plasma, liver, kidneys, and heart. Administration of a single IV dose of mRNA-3210 significantly lowered circulating Phe in PAH$^{enu2}$ mice. These findings are consistent with other reports demonstrating reductions in primary biomarkers of PA, MMA, and PKU in mice treated with mRNA therapeutics, with some results in line with the benefits received by standard-of-care treatments[4,21,22]. For example, the magnitude of plasma methylmalonic acid reductions in mRNA-3705-treated *Mut*$^{-/-}$;Tg$^{INS-CBA-G715V}$ mice were similar to the decreases in circulating

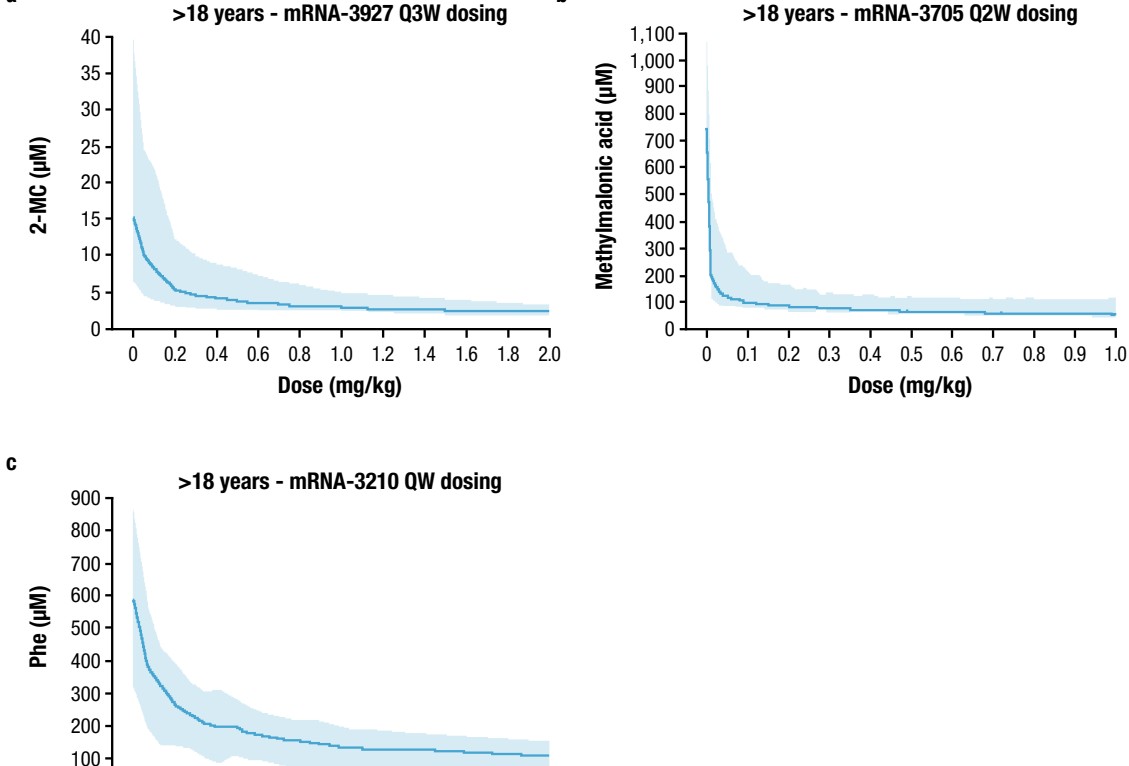

**Fig. 5 | Dose response for PA, MMA, and PKU in adult participants. a** Predicted relationship between mRNA-3927 dose and plasma 2-MC levels at steady state after Q3W dosing. **b** Predicted relationship between mRNA-3705 dose and plasma methylmalonic acid levels at steady state after Q2W dosing. **c** Predicted relationship between mRNA-3210 dose and blood Phe levels at steady state after weekly dosing. Solid lines represent the pre-dose levels at steady state (median). Shaded regions represent the 95% prediction interval. 2-MC 2-methylcitrate, MMA methylmalonic acidemia, mRNA messenger RNA, PA propionic acidemia, Phe phenylalanine, PKU phenylketonuria, Q2W every 2 weeks, Q3W every 3 weeks, QW every week. Source data are provided as a Source Data file.

**Table. 1 | Estimated FIH dose parameters for mRNA-3927, mRNA-3705, and mRNA-3210 based on PK/PD modeling**

|  | mRNA-3927 | mRNA-3705 | mRNA-3210 |
|---|---|---|---|
| LNP formulation | SM-86/OL-56 | SM-86/OL-56 | SM-86/OL-56 |
| Starting dose (mg/kg) | 0.3 | 0.1 | 0.4 |
| FIH dose species | *Pcca*⁻/⁻ (A138T) mouse, Sprague-Dawley rat, cynomolgus monkey | CD1 and *Mut*⁻/⁻;Tg^INS-CBA-G715V mouse, Sprague-Dawley rat, cynomolgus monkey | C57BL/6 B6.BTRB-*PAH*^enu2+/+ (wild-type) and PAH^enu2 mouse, cynomolgus monkey |
| NOAEL (mg/kg) | 3 | 5 | 3 |
| Safety relevant species | Sprague-Dawley rat | Sprague-Dawley rat | Sprague-Dawley rat |
| Safety margin | 10 | 50 | 7.5 |

*FIH first-in-human, LNP lipid nanoparticle, mRNA messenger RNA, NOAEL no-observed-adverse-effect-limit, PD pharmacodynamic, PK pharmacokinetic.*

methylmalonic acid concentrations observed following liver transplant in patients with MMA[23,24]. Interestingly, the biochemical responses observed in *Pcca*⁻/⁻ (A138T) mice treated with mRNA-3927 persisted for 3–4 weeks, while those observed in mRNA-3210 treated PAH^enu2 mice persisted for <168 h. Since the biochemical response for an mRNA therapeutic is driven largely by the half-life of the protein(s) for which it encodes, the fact that the PCCA/PCCB complex resides in the mitochondria relative to PAH, which resides in cytosol, may provide protection from proteolytic cleavage and factor into the longevity of the therapeutic effect. Collectively, the results presented herein, as well as the composite of published data related to mRNA therapeutics for PA, MMA, and PKU in mouse models, demonstrate that mRNAs delivered via LNPs can drive protein expression, produce enzyme activity in the pertinent tissue compartments, and reduce concentrations of the primary biomarkers of disease[4,21,22].

With this knowledge in hand, PK/PD modeling was used to determine the relationship between mRNA exposure and biomarker suppression for the 3 mRNA programs to guide starting doses and dosing regimens for FIH trials. A previously reported translational semimechanistic PK/PD model of mRNA-3927 was shown to adequately describe the plasma PK of PCCA/B mRNA in mice, rats, and monkeys. Herein, similar models were developed for MMA and PKU[3]. Allometric scaling of PK parameters for mRNA-3927, mRNA-3705, and mRNA-3210 indicated that 0.3 mg/kg every 3-weeks, 0.1 mg/kg every 3-weeks, and 0.4 mg/kg/week, respectively, were associated with initial start of plateauing in 2-MC, methylmalonic acid, and Phe levels. These findings, which are based on studies in murine models, have strong translation implications, as they mitigate the reliance on nonhuman primates for LNP-based therapeutics, paving the way for more ethically sound and efficient drug development processes. Furthermore, this integrated approach demonstrated consistency and relatively similar starting FIH dose levels across three different rare disease programs. As such, this approach holds promise in enhancing the accuracy of predicting FIH doses, ensuring optimal efficacy, and facilitating the rational design of clinical studies, ultimately promoting the likelihood of success in the clinical setting. As such, a starting dose of 0.3 mg/kg was selected for the FIH phase 1/2 study evaluating mRNA-3927 in patients with PA aged ≥1 year (ClinicalTrials.gov identifier: NCT04159103). Clinical trials assessing mRNA-3705 are currently ongoing; clinical trials assessing mRNA-3210 are planned.

In conclusion, these data demonstrate that mRNA therapeutics delivered via LNPs can reduce biomarkers for disease in murine models, and that PK/PD models based on these findings can be used to support the selection of efficacious doses for FIH studies. Overall, these investigational mRNA therapeutics show promise for providing clinical benefits in serious, metabolic disorders with no approved therapies and few or cumbersome treatment options.

## Methods
### Lipid nanoparticle production and formulation
mRNAs were synthesized and formulated by Moderna, Inc (Cambridge, MA, USA). Briefly, complete *N1*-methylpseudouridine

substituted mRNA encoding hPCCA and hPCCB proteins (mRNA-3927), hMUT protein (MMA-3705), or hPAH protein (mRNA-3210) was synthesized in vitro by T7 RNA polymerase–mediated transcription from a linearized DNA template containing the 5′ and 3′ untranslated regions and a poly(A) tail, as previously described[25,26]. After purification, mRNA was diluted in citrate buffer to the desired concentration and frozen. Lipid nanoparticles (LNPs) were formulated as described[4,21,26]. To generate LNP formulations for in vivo intravenous (IV) delivery, lipid components (SM-86, distearoylphosphatidylcholine, cholesterol, and OL-56 [polyethylene glycol-lipid conjugate]) were dissolved and the lipid mixture was combined with 25 mM sodium acetate buffer (pH 5) containing mRNA at a 3:1 (aqueous:ethanol) ratio using a microfluidic mixer (NanoAssemblr™, Precision Nanosystems, Vancouver, BC). Each mRNA-therapeutic IV unit contained the indicated dose of total RNA in the LNP dispersion formulated in 20 mM Tris (pH 7.5), 8% sucrose, and 1 mM diethylenetriaminepentaacetic acid. mRNA-3927 formulated LNPs were 70 nm in diameter, had a polydispersity index of 0.14, and were >96% encapsulated. mRNA-3705 formulated LNPs were 71 nm in diameter, had a polydispersity index of 0.07, and were 97% encapsulated. mRNA-3210 formulated LNPs were 77 nm in diameter, had a polydispersity index of 0.13, and were 95% encapsulated.

### Model selection
Propionic acidemia (PA) pharmacokinetic and pharmacodynamic (PK/PD) studies were conducted in *Pcca*⁻/⁻ (A138T) mice, which mimic the underlying causative propionyl-coenzyme A (CoA) carboxylase (PCC) enzyme deficiency, metabolic disturbances, and some clinical characteristics observed in patients with PA. *Pcca*⁻/⁻ (A138T) mice ubiquitously express a mutation in the human PCC α subunit protein (PCCA) described in patients with PA (p.A138T) that results in ~2% of normal PCC enzyme activity in the liver. These mice display marked elevations of toxic metabolites that directly accumulate due to PCC enzyme deficiency, including 2-methylcitrate (2-MC) and 3-hydroxypropionate (3-HP), as well as increases in the ratio of propionylcarnitine to acetylcarnitine (C3/C2). *Pcca*⁻/⁻ (A138T) mice additionally exhibit hyperammonemia consistent with patients with PA. Female *Pcca*⁻/⁻ (A138T) mice display higher biomarker levels than male mice do[27], although the reason for this is unclear. This discrepancy in biomarker levels between sexes has not been described for patients with PA.

*Pcca*⁻/⁻ (A138T) mice also have a diminished lifespan. Guenzel et al. reported a survival rate of ~75% at 3 months of age in *Pcca*⁻/⁻ (A138T) mice[28]. While the cause of death is unknown in these mice, PA-related complications are suspected. Herein, a lower mortality rate of 1.9% (7/374) in the *Pcca*⁻/⁻ (A138T) mouse colony was observed over a period of 5 months. This mouse model of PA does not exhibit the metabolic decompensations or neurologic manifestations observed in patients with PA. No other suitable animal models exist for in vivo pharmacology evaluation of mRNA-3927. Furthermore, there is no suitable animal model of PCCB deficiency.

For the MMA PD study, the *Mut*⁻/⁻;Tg^INS-CBA-G715V mouse model was used. *Mut*⁻/⁻;Tg^INS-CBA-G715V transgenic mice ubiquitously expresses a

mouse orthologue (p.G715V) of a well-described mutation (p.G717V) in MMA patients. At 5 weeks of age, these mice display an -26% reduction in body weight compared to heterozygous littermates[4]. Additionally, $Mut^{-/-}$;Tg$^{INS-CBA-G715V}$ mice exhibit a partial reduction in MUT activity in the liver which results in a moderate increase of plasma methylmalonic acid levels, akin to those observed in MMA patients with $mut^-$ who carry the MUT p.G717V mutation[4].

PAH$^{enu2}$ mice were used for the phenylketonuria (PKU) PK/PD studies. The PAH$^{enu2}$ mice are a model of heritable PKU created by ethylnitrosourea germline mutagenesis[29]. These mice carry a point mutation in exon 7 (c.835T>C; p.F263S) that results in <1% of wild-type hepatic phenylalanine hydroxylase (PAH) activity and a 15-fold increase in blood phenylalanine levels[30]. This model also recapitulates many of phenotypic aspects of PKU, including hyperphenylalaninemia[29], neurologic dysfunction[31], and catecholaminergic defects[32]. Similar to PKU humans, PAH$^{enu2}$ mice also display behavioral abnormalities and hypopigmentation, as early as 2 weeks of age[30].

## Study design and treatment groups

Multiple independent nonclinical studies evaluating the PK (mRNA concentrations) and PD (protein levels, enzyme activity, and biomarker levels) of mRNA-3927, mRNA-3705, and mRNA-3210 were conducted in murine models of PA, MMA, and PKU, respectively. For the PA PK study, $Pcca^{-/-}$ (A138T) mice (Jackson Laboratories, Bar Harbor, ME) aged 5.5 to 6 months received a single IV dose of PBS (control) or hPCCA + hPCCB mRNA (1:1 molar ratio) formulated in an LNP containing SM86/OL-56 (mRNA-3927) at 1.0 and 2.0 mg/kg via the tail vein (control: $n = 6$ mice/group; mRNA-3927: $n = 18$ mice/group). For the PA PD study, $Pcca^{-/-}$ (A138T) mice (Jackson Laboratories) aged 8 to 10 weeks received 2 IV bolus doses of Tris-sucrose (control) or mRNA-3927 at 0.5 mg/kg on Days 0 and 28 via the tail vein ($n = 6$ mice/group).

In the MMA PK study, CD1 (ICR) mice (Charles River Kingston, New York, NY) at least 8 weeks of age were administered a single IV dose of mRNA-3705 at 0.1, 0.5, or 1.0 mg/kg (12 males and 12 females/group). In the MMA PD study, groups of $Mut^{-/-}$;Tg$^{INS-CBA-G715V}$ mice aged 5–11 weeks (construction and analysis of mice as described[4,33]) were administered a single IV injection of phosphate-buffered saline (2 males, 2 females), mRNA-3705 at 0.2 mg/kg (3 males, 2 females), or mRNA-3705 at 0.5 mg/kg (2 males, 3 females). For the PKU PK/PD study, PAH$^{enu2}$ mice aged 11–17 weeks (Jackson Laboratories) were administered single IV doses of mRNA-3210, encoding human PAH at 0.25, 0.5, and 1.0 mg/kg ($n = 6–16$/treatment group) or 0.3 mL of 0.9% sodium chloride as control. All doses (mg/kg) were based on mRNA concentrations only and did not consider the molecular weight of the LNP. Target temperatures of 68° Fahrenheit (F) to 79°F with a relative target humidity of 30–70% were maintained. A 12-h light/12-h dark cycle was maintained. Certified Rodent ProLab® Rodent Diet 5P76 IsoPro RMH 3000 was provided ad libitum. Water was available ad libitum to each animal via an automatic watering system.

All animal experiments involving $Pcca^{-/-}$ (A138T) mice or $Mut^{-/-}$;Tg$^{INS-CBA-G715V}$ mice were approved and conducted in accordance with regulations from the ModernaTX, Inc. Institutional Animal Care and Use Committee (IACUC). The research reporting of all animal experiments abide by the ARRIVE guidelines. Protocols and amendments or procedures involving CD1 mice were reviewed and approved by Charles River Laboratories IACUC (Shrewsbury, MA). Female mice were socially housed and male mice were single-housed. For studies in $Pcca^{-/-}$ (A138T) mice or $Mut^{-/-}$;Tg$^{INS-CBA-G715V}$ mice, the welfare of animals was monitored by viability checks, clinical observations and individual body weight measurements daily. Animals were euthanized by carbon dioxide asphyxiation followed by exsanguination. For studies with CD1 mice at Charles River Laboratories, the welfare of animals was monitored by twice daily mortality/moribund checks, daily cage side observations, post-dose and detailed clinical observations and individual body weight measurements at least biweekly. At scheduled

termination, animals were anesthetized with isoflurane and underwent half-body perfusion (abdominal aorta clamped above the needle) with sodium chloride 0.9% (saline), heparin (1000 IU/mL), 1% sodium nitrite, and then with chilled PBS. The animal body was flushed until the solution that is draining the atrium was considered clear by visual inspection. Target temperatures of 68° Fahrenheit (F) to 79°F with a relative target humidity of 30–70% were maintained. A 12-h light/12-h dark cycle was maintained. Block Lab Diet® (Certified Rodent Diet #5002, PMI Nutrition International, Inc) was provided ad libitum. Water was available ad libitum to each animal via an automatic watering system.

Study plans, amendments, and procedures involving PAH$^{enu2}$ mice were reviewed and approved by Charles River Laboratories IACUC (Montreal ULC, Sherbrooke Site). During the study, the care and use of PAH$^{enu2}$ mice was conducted with guidance from the USA National Research Council and the Canadian Council on Animal Care. The research reporting of all animal experiments abide by the ARRIVE guidelines. Female mice were socially housed and male mice were single-housed, and both female and male mice were provided with items such as a hiding device, chewing object, except during study procedures and activities. The welfare of animals was monitored by twice daily mortality/moribund checks, daily cage side observations, post-dose and detailed clinical observations and individual body weight measurements at least biweekly. At scheduled termination, animals were anesthetized with isoflurane and underwent half-body perfusion (abdominal aorta clamped above the needle) with sodium chloride 0.9% (saline), heparin (1000 IU/mL), 1% sodium nitrite, and then with chilled PBS. The animal body was flushed until the solution that is draining the atrium was considered clear by visual inspection. Target temperatures of $22 \pm 3°$ Celsius with a relative target humidity of 30–70% were maintained. A 12-h light/12-h dark cycle was maintained. Lab Diet Certified CR Rodent Diet 5CR4 was provided ad libitum. Water was available ad libitum to each animal via an automatic watering system.

## mRNA quantification

For the PA and MMA PK studies, mRNA from mRNA-3927 or mRNA-3705 was quantified by branched DNA analysis using the QuantiGene TM Singleplex assay kit according to the manufacturer's instructions (Thermo Fisher Scientific, Waltham, MA). For the PA PK study, blood was collected at 0, 0.5, 1, 2, 4, 6, 8, 10, 24, and 48 h via submaxillary bleeding (0.15 mL/time point; 6 mice/time point) and dispensed into tubes containing dipotassium ethylenediaminetetraacetic acid and centrifuged at 4 °C for 10 min at $1300 \times g$, and the supernatant was collected as plasma. For the MMA PK study, blood was collected at 0, 0.5, 1, 2, 4, 6, 8, 10, 24, and 48 h via venipuncture (0.1 mL/time point; 6 mice/time point) and dispensed into tubes containing dipotassium ethylenediaminetetraacetic acid. Within 15 min of collection, samples were centrifuged at 4 °C for 15 min at $1300–2000 \times g$, and the supernatant was collected as plasma. For the PA study, the lower limit of quantification (LLOQ) for hPCCA and hPCCB mRNA was 0.225 and 0.664 ng/mL, respectively. PK parameters, including AUC$_{tlast}$, apparent clearance rate, and volume of distribution, were assessed.

In the PKU study, liver samples were collected for mRNA and protein quantification at 2, 8, 24, 48, 86, and 168 h after treatment. Briefly, following isoflurane anesthesia, the mice underwent half-body perfusion (abdominal aorta clamped above the needle) with 1000 IU/mL heparin in 0.9% sodium chloride and 1% sodium nitrite, followed by 1X PBS (4 °C). The animal body was flushed until the solution draining the atrium was deemed clear by visual inspection. Liver samples from 2 mice (15–25 mg each) were collected, rinsed in 1X PBS, dried with sterile drape, weighed (wet weight), and placed into RNAase-free tubes for mRNA analysis. Blood samples for mRNA quantification were collected at baseline and at 0.25, 0.5, 1, 2, 4, 6, 8, 10, 12, 24, 48, 96, and 168 h postdose. Samples were centrifuged at $2200 \times g$ for 10 min at

4 °C, and the resultant serum was separated. mRNA from mRNA-3210 was quantified in both liver and serum samples by real-time reverse transcription-quantitative polymerase chain reaction (PCR) assay using the QuantStudio™ Real-Time PCR System (Thermo Fisher Scientific). The LLOQ was 0.000000179 ng/mg in liver samples and 0.00000761 ng/mL in serum samples.

## Protein concentration assays

For the MMA PD study, of $Mut^{-/-}$;Tg$^{INS-CBA-G715V}$ mice were euthanized 24 h after treatment by carbon dioxide asphyxiation. Liver samples were collected, snap frozen, and stored at ≤80 °C. Livers were homogenized in 100 mM ammonium bicarbonate buffer with 8 M urea.

MUT isotopically labeled signature peptide (human MUT-specific peptide, IIADIFEYTAK*, in which natural carbon and nitrogen atoms on lysine were fully replaced with $^{13}$C and $^{15}$N isotopes, respectively; Thermo Pierce, Rockford, IL) were used as internal standards and spiked into each sample. Liver proteins were denatured, followed by reduction using 5 mM Tris (2-carboxyethyl) phosphine hydrochloride (75259, Sigma-Aldrich Inc, St. Louis, MO) at 37 °C for 1 h, alkylation with 10 mM iodoacetamide (I6125, Sigma-Aldrich Inc) at 25 °C in the dark, and then digestion at 37 °C for 15 h with trypsin (trypsin:protein = 1:50 w/w; Catalog #V5280, Promega Inc, Madison, WI); digestion was stopped by adding formic acid (28905, Thermo Fisher Scientific, Waltham, MA). Samples were desalted on the SOLA plate (60309-001, Thermo Fisher Scientific), dried, and resuspended in water with 2% acetonitrile and 0.1% formic acid (A955-1, LS120-1, Thermo Fisher Scientific, Waltham, MA). Total protein (0.25 µg) was loaded onto the column (75 µm × 15 cm column packed with Waters Acquity ethylene-bridged hybrid resin 1.7 µm × 130 Å) and subjected to a liquid chromatography-tandem mass spectrometry analysis (Thermo Easy 1000 nano-Ultra-Performance Liquid Chromatography (UPLC), Orbitrap Fusion Mass Spectrometer, Thermo Fisher Scientific). Water (A) and acetonitrile (B) with 0.1% formic acid were used for liquid chromatography separation at a flow rate of 300 nL/min. The gradient was as follows: 2% B to 7% B in 5 min; 7% to 35% B in 45 min; 35% to 80% B in 5 min; 80% to 95% B in 2 min; and 95% B for 5 min. The mass spectrometry was performed in continuous parallel reaction monitoring mode: spray voltage, 1900 V; S-lens radio frequency, 60%; isolation width, 1.4 m/z; higher-energy collisional dissociation, 30% collision energy; detector, orbitrap; resolution, 60,000; scan range, 100–1800 m/z; automatic gain control, 5E4; and maximum injection time, 200 ms. The LLOQ was 0.5 ng/mg protein, and values below the LLOQ were imputed as 0. A complete description of protein purification and LC-MS/MS can be found in the Supplementary Methods.

In the PKU PK/PD study, mice were euthanized, and liver samples were collected as described above. Approximately 100 mg of liver tissue was transferred to a 2 mL Precellys® soft tissue homogenizing CK14 lysis tube (Bertin Instruments, Rockville, MD). The tissue samples were homogenized by adding T-PER protein extraction reagent at a 3:1 ratio (300 µL reagent: 100 mg tissue). The tissue samples were processed using a Precellys® tissue homogenizer for 20 s at 6000 rotations per minute, 3 times. The tubes were centrifuged at 2350 × g for 5 min at 4 °C, after which the supernatant was collected as homogenate. Based on the total protein content determined using bicinchoninic acid analysis, the amount of tissue homogenate that would equal 400 µg of total protein was transferred to a 2 mL lo-bind 96-well plate. After adding enough 8 M urea:100 mM ammonium bicarbonate buffer to bring the total volume to 200 µL, 10 µL of 200 mM dithiothreitol (30.9 mg/mL) in water was added, and the samples were centrifuged. Using a liquid handling workstation (TOMTEC, Hamden, CT), samples were mixed by aspirating and dispensing for 10 cycles. The plate was allowed to sit at ambient temperature for 5 minutes, and then was placed in an oven at 37 °C for 60 min. After allowing the plate to sit for 15 min to return to ambient temperature, 20 µL of 250 mM iodoacetamide (46.2 mg/mL) in water was added, and the plate was

centrifuged. The samples were again mixed by aspirating and dispensing for 10 cycles, and then placed in the dark. The samples were allowed to react for 30 min, and then 20 µL of 1 mg/mL trypsin in water was added and the plate was centrifuged. The samples were again mixed by aspiration, allowed to digest overnight (approximately 16 h), vortex mixed, and then allowed to cool for 15 min at ambient temperature. After centrifuging, 10 µL of 10% formic acid in water were added to quench the reaction, and the plate was centrifuged again. The samples were mixed by aspiration, and the plate was centrifuged. The supernatants (100 µL) were transferred to a 2 mL lo-bind 96-well plate, and 10 µL of a 250/1250 ng/mL $^{13}$C$_6$- and $^{15}$N$_2$-labeled surrogate peptide LFEENDVNLTHIESRPSR (LFE) solution (prepared in 30/70 v/v acetonitrile/water) was added to each sample. The plate was centrifuged, and the samples were mixed by aspiration. Tissue homogenate samples were analyzed for LFE peptide over the range of 0.500–200 ng/mL. Extracts were analyzed using LC-MS/MS in positive ionization mode under optimized conditions for detection of LFE-positive ions formed by electrospray ionization, and hPAH concentrations were quantified.

## Enzyme activity assays

PCC enzyme activity was measured in the PA PD study using a radiometric activity assay as previously described[34]. Briefly, $Pcca^{-/-}$ (A138T) and wild-type mice were sacrificed 2 days after the second IV treatment on Day 30, and livers were harvested after mice were perfused with PBS to avoid blood contamination. Livers were homogenized in a homogenization buffer containing 10 mM Trizma® base-MOPS (3-[N-morpholino] propanesulfonic acid; Sigma-Aldrich), 1 mM Trizma® base ethylene glycol-bis (2-aminoethylether)-N,N,N',N'-tetraacetic acid (EGTA), and 200 mM sucrose supplemented with protease inhibitors. The liver homogenate was further processed to isolate mitochondria after 2 steps of centrifugation at 600 × g for 10 minutes and 7000 × g for 10 min sequentially. The resultant mitochondrial pellet was resuspended in mitochondrial lysis buffer containing 0.5% Triton X-100, 1 mM dithiothreitol, and 10 mM HEPES (4-(2-hydroxyethyl)-1-piperazineethanesulfonic acid) at pH 7.4, supplemented with protease inhibitors (Sigma-Aldrich). After 6 freeze/thaw cycles, the fraction was centrifuged at 18,000 × g for 15 min to obtain the mitochondrial matrix. Protein estimation was performed using the bicinchoninic acid (BCA) method (BCA Protein Assay Kit; Thermo Scientific Pierce, Rockford, IL) according to the manufacturer's recommendations. Hepatic mitochondrial fractions were mixed with PCC substrates (adenosine triphosphate, propionyl-CoA, and $^{14}$C radioisotope-labeled sodium bicarbonate). The reaction mixture was incubated at 37 °C for 15 min, 5% trichloroacetic acid was added to stop the reaction, and the solution was centrifuged at 13,000 × g for 5 min. The supernatant was collected for quantification of radioisotope-labeled methylmalonyl-CoA using a Microbeta2 scintillation counter (Perkin Elmer, Waltham, MA).

MUT activity was determined in the MMA PD study as previously described[35]. Hepatic lysates were incubated with 5'-deoxyadenosylcobalamin (200 µM; C0884, Sigma-Aldrich Inc, St. Louis, MO) and a racemic mix of methylmalonyl-CoA (1 mM; M1762, Sigma-Aldrich Inc) at 37 °C for 15 min. MUT enzyme reactions were terminated by the addition of 50 µL of 100 g/L trichloroacetic acid with vortexing. Samples were centrifuged at 13,000 × g for 5 min. Chromatographic separation and quantification were accomplished with high-performance liquid chromatography (HPLC). The supernatants (20 µL) were injected and separated on a Poroshell EC-C18 120 HPLC column (695975-302, Agilent Technologies, Santa Clara, CA) equilibrated with 100 mM acetic acid (A6283, Sigma Aldrich Inc) in 100 mM sodium phosphate (AC343815000, Thermo Fisher Scientific) buffer, pH 7.0 (Solvent A). Solvent B was prepared by the addition of 18% v/v methanol in Solvent A. Elution was performed with a linear methanol gradient: Solvent B increased from 0% to 95% over 0–15 min (95%

Solvent B) and then stayed at 95% from 15 to 25 min (95%) with a flow rate of 0.5 mL/min.

## Metabolite concentration assays

For the PA PD study, blood samples were collected prior to treatment (Day -6) and on Days 2, 8, 14, 22, and 28 following treatment via sub-mandibular bleeding (~60 μL/time point) and dispensed into prechilled tubes containing dipotassium ethylenediaminetetraacetic acid. Within 15 min of collection, samples were centrifuged at 4 °C for 10 min at 1300 × g, and the supernatant was collected as plasma. Plasma concentrations of 2-MC, C3, C2 (for normalization of C3), and 3-HP were quantified by LC-MS/MS as previously described (Charles River Laboratories, Worchester, MA)[36–38].

For the MMA PD study, 40 μL of blood was collected into sodium heparin tubes via submental bleed at baseline (Day -7) and 24 h post-dose. Blood samples were gently mixed and centrifuged (4 °C for 15 min at 3000 × g), and the supernatant was stored at ≤80 °C. Plasma methylmalonic acid was analyzed and quantified by LC-MS/MS as previously described[39]. The LLOQ was 5 or 10 μM, depending on the dilution; values below the LLOQ were imputed as 0. $Mut^{-/-}$;Tg$^{INS-CBA-G715V}$ mice were euthanized, as described above, and liver, kidney, and heart (100–200 mg) samples were collected, snap frozen, and stored at ≤80 °C. Tissue samples were homogenized by Omni Bead Ruptor (Omni International, Kennesaw, GA) following the addition of 4 equivalents (weight:volume) of 80:20 water:acetonitrile. Methylmalonic acid extracts were derivatized with BuOH/HCl to generate the corresponding butyl ester derivative and analyzed and quantified by LC-MS/MS, as previously described[39]. The LLOQ was 25 nmol/g tissue; values below the LLOQ were imputed as 0.

For the PKU PK/PD study, blood samples were collected via the saphenous or submental vein at baseline and 1, 12, 24, 48, 72, 96, and 168 h postdose. Blood was collected using a Mitra® Volumetric Absorptive Micosampling (VAMS) device according to manufacturer's instructions (Trajan, Torrance, CA). Briefly, the first drop of blood was discarded; the second drop of blood was applied to the tip of the VAMS device, avoiding contamination from the animal's skin. Once full, the VAMS device was placed into a drying rack and incubated at room temperature for ~24 h (maximum of 36 h); samples were stored at −20 °C. Phe was analyzed using an Agilent 1200 LC system and a Waters™ XBridge BEH HILIC column (Waters™, Milford, MA). The analytes were detected using a SCIEX API 5000™ Triple Quad LC-MS/MS system equipped with an ESI (TurboIonSpray ionization source operated in the positive ionization mode (SCIEX, Farmingham, MA).

## Approach to the FIH projections for mRNA-3927, mRNA-3705, and mRNA-3210

A 3-compartment population PK model of mRNA-3927/mRNA-3705 was developed. The model included compartments for liver uptake and elimination and was based on a published model in the literature for an LNP-formulated small interfering RNA therapeutic[40]. The model included fixed allometric weight-based scaling factors for clearance and volume of 0.75 and 1.0, respectively. For PD biomarkers, classic indirect response models with inhibition of kin were developed for 3-HP and 2-MC/MMA, incorporating PCC/hMUT enzyme-induced decreases in organic acid accumulation.

The datasets utilized for the mRNA-3927 PK/PD modeling have been previously reported and are listed in Supplementary Table 8[3]. mRNA-3705 PK/PD datasets relevant for development of the MMA PK/PD model were obtained from studies listed in Supplementary Table 9. No-observed-adverse-effect-limits represent the mid-dose levels observed in the safety studies.

A simple turnover model linking mRNA (2-compartment PK model) to Phe was sufficient to characterize the reduction of Phe over time. Final PK/PD parameters in the mouse PAH$^{enu2}$ disease model were allometrically scaled to humans with the target of suppressing Phe level to <360 μM. mRNA-3210 PK/PD studies from disease and wild-type mice that were used for model development, as well as cynomolgus monkey data used to qualify the scaling factor for extrapolation to higher species, are listed in Supplementary Table 10.

Model development used standard modeling tools (eg, Phoenix NLME or Monolix). Model evaluation was performed using appropriate diagnostics (eg, visual predictive check, goodness-of-fit, residual and Akaike information criterion). Data pre- and post-processing were performed using RStudio (version 4 and higher).

## Statistical analyses

In the PA PK study, descriptive statistics were generated using Phoenix, version 8.1. Data were expressed as means ± SD. In the PA PD study, statistical analyses were performed using GraphPad Prism, version 7.01. Data were expressed as means ± SD.

In the MMA PK study, computation of noncompartmental analysis, descriptive statistics, and ratios were performed using Phoenix, version 1.4. For the MMA PD study, statistical analyses and descriptive statistics were performed using GraphPad Prism, version 9. Data were expressed as means ± SD. For the plasma biomarker (methylmalonic acid), a repeated measures 2-way analysis of variance (ANOVA) was performed. For tissue biomarker concentrations, liver MUT protein concentrations, and liver protein activity, 1-way ANOVAs were performed. For significant ANOVA findings, Dunnett's pairwise comparison tests were performed between groups. Two-tailed $P$ values < 0.05 were considered statistically significant. AUC$_{tlast}$ was calculated for both the PA and MMA PK studies using the linear trapezoidal method.

For the PKU PK/PD study, computation of non-compartmental analysis, descriptive statistics, and ratios were performed using Phoenix, version 8.3. Concentration data were expressed as mean ± SD.

## Reporting summary

Further information on research design is available in the Nature Portfolio Reporting Summary linked to this article.

## Data availability

The datasets generated during and/or analyzed during the current study are not publicly available due to the propriety nature of the LNP therapeutics described herein. Access to data and supporting documents from qualified external researchers may be available upon request. Source data for presented figures are provided with this paper. Source data are provided with this paper.

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

## Acknowledgements

Medical writing and editorial support were provided by Laura Watts, PhD, of Lumanity Communications Inc., and were funded by Moderna, Inc.

## Author contributions

R.B., K.C. and L.J. contributed equally to this work. R.B. participated in the design of the studies, protocol writing, interpretation of data, and writing the report for the PKU study. K.C. participated in the design of the studies, protocol writing, analyzing and interpreting data, and writing the report for the MMA study. L.J. participated in the design of the studies, protocol writing, data analysis and interpretation, and writing the report for the PA study. M.L. analyzed and interpreted data. H.S. provided bioanalytical guidance, participated in reviewing study reports, and interpreted data. H.Z. participated in reviewing and interpreting data, and reviewing the report for the PKU study. L.C. participated in the design of the studies, protocol writing, data interpretation, writing the report for the PKU PK/PD studies, and interpretion of PK data for the PA and MMA studies. N.K. participated in analyzing and interpreting data. I.L.R. performed PKU translational PK/PD modeling. L.V. participated in reviewing and interpreting data, drafting of PK/PD report for the PKU

study, and reviewing for the PA and MMA studies. R.D. participated in the design of the studies, interpreting data, and writing the report for the PKU study. A.C. participated in the design of the studies, executing experiments, and analyzing data. L.Y. participated in the design of the PA studies, conducting experiments, processing samples, and generating data. L.R. made substantial contributions to the conception and design of the work and technology used for the PKU study. A.F. participated in the design of the studies, protocol writing, execution of the studies, data interpretation for the PKU and MMA studies, and design and protocol writing for the PA study. L.G. provided guidance on experimental design and data interpretation. P.F. provided guidance on experimental design and data interpretation. P.G.M. created the PA, MMA, and PKU projects for the Sponsor and guided the ideas for the execution of the work described. All authors reviewed the manuscript, approved the final version, and vouch for data accuracy and completeness.

## Competing interests

All authors are employees of Moderna, Inc., and may hold stock/options in the company.
