## [Peer Review File · Nature Communications]

REVIEWER COMMENTS

Reviewer #1 Comments:

1. Summary of key results: This work builds upon previous research from the authors showing that LNP-mediated hepatic delivery of therapeutic mRNA can lead to improvements in biomarkers of two metabolic diseases in murine models: PA and MMA. Baek et al. provide a focused report performing PK/PD analysis of different dosages of three mRNA-LNP therapies in mouse models for rare genetic metabolic diseases (PA, MMA, PKU). The authors take these results and use existing modeling approaches to predict recommended dosages for first-in-human clinical trials.

2. Originality and significance: This work adds new standard PK/PD analysis for varying different dosages of three mRNA LNP therapies. However, these same therapies have already been studied in the same mouse disease models using the same biomarkers as a readout before to show therapeutic efficacy *in vivo*. Two of these therapies were in published studies from this research group (An et al., Cell Rep 2017; Jiang et al., Nat Comm 2020) and one from another group (Perez-Garcia et al., Mol Ther Nucleic Acids 2022). This substantially reduces the originality and significance of this work.

The primary original findings of this work are: 1) adding the PK/PD analysis of 2-3 different dosages of a single dose in these same animal models; and 2) offering specific recommended dosages for first-in-human therapeutic trials. There is a related finding in the co-submitted paper (Grunewald, et al., referenced in this work) indicating the recommended dose for PA treatment was used in a FIH clinical trial and appears to be efficacious and not prohibitively toxic using data from the trial still in progress. This offers evidence that the predictive doses can be significantly relevant to translation to the clinic. However, it also indicates the findings for the predicted PA therapeutic doses in this manuscript were already employed in an approved clinical trial started over a year ago.

3. Data and methodology: The overall approach is straightforward and sound. Dosage of each of the therapies is changed and mRNA levels, protein levels, and biomarker levels are tested over time to show efficacy and dose responsiveness. One concern is the lack of control subjects in many of the experiments. While there is often a dose-dependent response, it is difficult to draw conclusions about efficacy and the significance of changes without this. Data are clearly presented.

The specific animal models chosen should be at least briefly rationalized. There are multiple mouse models for MMA, why choose a specific hypomorphic allele? Why inject both PCCB and PCCA mRNA into the *pccb* ^{-/-} hypomorphic mutant? Neonatal lethality? Similarity to patient genotypes? Experimental design rationale would help the reader here.

4. Appropriate use of statistics: Results of statistical analysis and significance are not clear or lacking. The experiments shown in Fig. 3 (for hMUT) are the only ones that have indicated relative p-values. If the data in the paper are largely not significant (either between dosages or compared to controls), it would

make drawing conclusions and proposing doses for FIH studies problematic. If much of the data are significant, indications of significance should be made on their respective graphs and p-values provided in the text, legends, or supplementary data. Alternatively, if AUC values are more fitting for some datasets, those values should be included, compared, and discussed. While some of the statistical methodology is discussed in methods, it is absent from most of the results.

5. Conclusions: The conclusions presented are that “these data demonstrate that mRNA therapeutics delivered via LNPs can reduce biomarkers for disease in murine models, and that PK/PD models based on these findings can be used to support the selection of efficacious doses for FIH studies.” The first conclusion is valid, however, this conclusion was already shown in each of the separate publications indicated earlier. This current study validates that previous work but it also uses the same therapies, animal models, and biomarkers employed previously. In that sense, this conclusion is robust and reliable and is consistent with previous literature. The conclusion regarding selection of proper FIH dosing has proven to be valid for one of the three therapies, however, to what extent these dosing predictions are effective in clinical patients for the other two therapies remains to be seen. To that end, it is difficult to say if the second batch of conclusions are robust or reliable.

6. Suggested improvements: Two major areas of improvements would help this manuscript.

Statistical analysis:

-Add p-values for experiments in legend, graphs, or in text. Indicate and discuss statistical significance, or lack thereof, of findings. Indicate significance on all relevant graphs.

- Alternatively, if AUC values are more fitting for some datasets, those values should be included, compared, and discussed.

-Perform experiments for PKU mice (Fig. 4A-C) with controls. There is a control group in Fig. 4D (blood Phe levels) but none of the others in this set of experiments.

-Compare dosing regimens. This work examines the concentration of LNP-mRNA delivered but previous preclinical studies (cited above) with these therapeutics show that repeat administration (once per week for several weeks) maintains improvements in biomarker readouts without major toxicity. For at least one or two of the dosages per therapeutic, what do the PK/PD results look like when comparing a single dose administered a week apart vs. another regimen (once every three days, for example)? This is more reflective of clinical administration in patients and better informs FIH dosing recommendations.

-Multiple times AUC(tlast) results are mentioned, but no values are provided. (p. 6 line 107, p. 7 line 126-139)

Broader significance/impact: While the clinical relevance is clear, does this provide broader insights into LNP-mediated mRNA therapies, improve upon or alter our methodologies for predicting dosing, or substantially expand our understanding of these investigational therapies? To address this:

-If there is a broader impact from this work or relevance to an array of interests, clearer explanation of this should be made in the introduction and discussion.

-If there is a further refinement of study or approach using these specific therapies, that should be discussed.

7. References: In some cases, original research should be cited. For example, a review from Chandler and Venditti (Hum Gene Ther 2019) is cited as both evidence of the advantages of LNP in terms of its intracellular transport and genotoxicity and evidence of toxicity of viral gene delivery. However, this citation is a review with a sentence or two briefly stating those conclusions. Original literature or a more specific, comprehensive review should be cited.

8. Clarity and context: This study is very clear and focused. The goals and conclusions from the introduction through the results and discussion are consistent and clear. More discussion of the broader scientific or clinical significance of this work in the introduction and discussion would benefit the reader in contextualizing this work.

Reviewer #2 Comments:

The manuscript describes the outcomes of treating murine models of propionic acidemia, methylmalonic acidemia, and phenylketonuria with mRNA delivered with liver-directed lipid nanoparticle (LNP) technology.

Although the experimental details differ for the three individual models, the methods are sound. The manuscript is well-written. The preclinical data on the propionic acidemia model formed the rationale for a first in human clinical trial. However, the designs of the individual preclinical experiments for the three diseases are rather different. Although the common root is the investigation of mRNA therapy in inborn errors of metabolism, given that there are distinct biological and clinical differences between the disorders, the rationale for lumping all three experiments into a single manuscript is unclear.

To facilitate comparisons to other investigators in the field, more details regarding the design, production, and purification of the mRNA/LNP formulation should be provided.

Regarding the mRNAs, how were these synthesized? Did they incorporate modified bases and optimized untranslated regions? How were they capped? How were they purified prior to loading into LNPs?

Regarding the LNPs, the terminology SM86/OL-56 is used in the manuscript to identify the lipids incorporated into the LNP, but this will have no meaning to anyone who is not an aficionado of the field. The meaning of these abbreviations should be included. However, more information regarding the LNP structure is necessary. What were the final ratios of the cationic lipid: phospholipid: cholesterol: PEG? What method was used to mix the lipids with the mRNA? How were the complexes then purified? In what solution were the complexes finally suspended for intravenous injection?

I understand that some of the details of the LNP structure may be proprietary, but I am not asking for sufficient information that reagents can be duplicated. However, the details I have requested, I propose, are key to understanding both efficacy and toxicity of mRNA/LNP therapeutics. Without this information, it is impossible to compare the outcomes of different experiments in the literature.

Also, this is a small point, but an irritation I have with the mRNA field as a whole. I recommend clarifying at the outset that the calculated dose given to the mouse is the amount of mRNA in mg/kg body weight delivered and does not include the mass of the LNP. Then, the molar ratio of mRNA to lipid should also be given.

Now turning to the results of the specific experiments:

1. Propionic acidemia

The PCCA deficient models harbors a missense variant that apparently retains some PCC expression. Is it known whether the human PCCB monomer can complex with endogenous mutant mouse PCCA monomer? Or whether the human PCCA can complex with mouse PCCB? If so, this could alter the effectiveness of the therapy if one can deliver only a partial amount of message in comparison to the amount of native mRNA and PCC protein already present.

The treatment did yield an increase in liver PCC activity in the PCC^{-/-} mice over control-treated mice. How does this activity compare to that treated in wild type PCC^{+/+} mice?

It is intriguing that the duration of biochemical response to therapy was 3-4 weeks in the PCC^{-/-} model but less than one week for PKU. Do the authors have an explanation for this?

2. Methylmalonic acidemia

Again, was the liver MUT enzyme activity in treated mice compared to that of wild type mice?

This experiment was apparently terminated at 24 hours. Was no attempt made to study the duration of the treatment response?

3. Phenylketonuria

I note that hPAH mRNA was detectable in blood for much longer in this experiment than was the case for mRNA in the propionic acidemia experiment. Is there a true biological difference here or this a measurement artefact due to the differences in the sensitivity in the methods used to detect mRNA in the two experiments?

24 hours post dose, the mean concentration of hPAH mRNA measured in liver was 51.57 ng/g. First, one assumes this is ng/g liver wet weight but I could not find the definition.

If the units are ng/g liver tissue, then I calculate that approximately 0.5% of the delivered mRNA dose remains in the liver at 24 hours. Is the fate of the other 99.5% of the delivered dose known?

How does this amount of hPAH mRNA compare to native mPAH mRNA expression?

After mRNA delivery, hPAH protein production clearly increased and blood phenylalanine concentrations decreased demonstrating that the protein had PAH activity, but were any attempts made to actually measure enzymatic PAH activity in liver homogenates from the treated PKU mice?

4. PK/PD modeling

Why do the no-observed-effect-limits differ between the different disease models? How were these determined?

5. Conclusions

The ultimate goal of all three clinical projects was ultimately to determine an ideal path forward to clinical trial for each mRNA therapeutic. The PK/PD modeling providing idealized dose estimates for each mRNA, but the manuscript does not mention the predicted necessary dosing frequency. Perhaps this can

ultimately only be optimized in humans, but certainly these experiments must suggest a likely initial necessary dosing frequency.

Characterizing the Mechanism of Action for mRNA Therapeutics for the Treatment of Propionic Acidemia, Phenylketonuria, and Methylmalonic Acidemia

(Reference Number: NCOMMS-23-49825-T)

Comments from *Nature Communications*

Please see below for a grid with a summation of the comments and annotations for line numbers corresponding to our revisions in the *tracked version* of the manuscript.

Comment Number	Comments	Author Response and Changes Made	Page
Reviewer #1 Comments:			
1	Summary of key results: This work builds upon previous research from the authors showing that LNP-mediated hepatic delivery of therapeutic mRNA can lead to improvements in biomarkers of two metabolic diseases in murine models: PA and MMA. Baek et al. provide a focused report performing PK/PD analysis of different dosages of three mRNA-LNP therapies in mouse models for rare genetic metabolic diseases (PA, MMA, PKU). The authors take these results and use existing modeling approaches to predict recommended dosages for first-in-human clinical trials.	We thank the reviewer for their thorough summary of the manuscript.	NA
2	Originality and significance: This work adds new standard PK/PD analysis for varying different dosages of three mRNA LNP therapies. However, these same therapies have already been studied in the same mouse disease models using the same biomarkers as a readout before to show therapeutic efficacy in vivo. Two of these therapies were in published studies from this research group (An et al., Cell Rep 2017; Jiang et al., Nat Comm 2020) and one from another group (Perez-Garcia et al., Mol Ther Nucleic Acids 2022). This substantially reduces the originality and significance of this work. The primary original findings of this work are: 1) adding the PK/PD analysis of 2-3 different dosages of a single dose in these same animal models; and 2) offering specific recommended dosages for first-in-human therapeutic trials. There is a related finding in the co-submitted paper (Grunewald, et al., referenced in this work) indicating the recommended dose for PA treatment was used in a FIH clinical trial and appears to be efficacious and not prohibitively toxic using data from the trial still in progress. This offers evidence that the predictive doses can be significantly relevant to translation to the clinic. However, it also	Language has been added to the Discussion of the manuscript highlighting the strong translational implications of these data. For example, nonhuman primate models are often employed in later preclinical phases when assesses the safety and efficacy of mRNA-lipid nanoparticle therapeutics. Here, we demonstrate that integrating murine models with PK/PD modeling shows promise in accurately predicting dosing regimens for FIH clinical trials, possibly mitigating the reliance on nonhuman primates. These findings, which are based on studies in murine models, have strong translation implications, as they mitigate the reliance on nonhuman primates for LNP-based therapeutics, paving the way for more ethically sound and efficient drug development processes. Furthermore, this integrated approach demonstrated consistency and relatively similar starting FIH dose levels across three different rare disease programs. As such, this approach holds promise in enhancing the accuracy of predicting FIH doses, ensuring optimal efficacy, and facilitating the rational design of clinical	P12, Ln 244-251

	indicates the findings for the predicted PA therapeutic doses in this manuscript were already employed in an approved clinical trial started over a year ago.	studies, ultimately promoting the likelihood of success in the clinical setting.									
3	Data and methodology: The overall approach is straightforward and sound. Dosage of each of the therapies is changed and mRNA levels, protein levels, and biomarker levels are tested over time to show efficacy and dose responsiveness. One concern is the lack of control subjects in many of the experiments. While there is often a dose-dependent response, it is difficult to draw conclusions about efficacy and the significance of changes without this. Data are clearly presented. The specific animal models chosen should be at least briefly rationalized. There are multiple mouse models for MMA, why choose a specific hypomorphic allele? Why inject both PCCB and PCCA mRNA into the pccb^{-/-}-hypomorphic mutant? Neonatal lethality? Similarity to patient genotypes? Experimental design rationale would help the reader here.	In our studies, we include untreated or vehicle-treated disease animals as a control to evaluate impact of the mRNA-therapy in a disease background. We did not administer our mRNA therapy to unaffected wild-type (WT) or heterozygous mice in these experiments. mRNA analysis was specific to the mRNA test article. Therefore, no test article specific mRNA would be evaluable in WT mice. The assay to measure the mRNA-derived protein is specific to the human protein and therefore is not evaluable in WT mice. We have evaluated the biomarker levels in WT or heterozygous mice in separate studies that have been previously published for PA and MMA. In general, the biomarker levels in WT levels are not detectable or below the limit of assay quantification. PCC activity in WT mice has been added to Figure 2c for comparison.   <caption>Liver PCC protein activity (nmol/min/mg protein)</caption>   Group Activity (nmol/min/mg protein)     Control ~2, ~3, ~4, ~5, ~6   0.5 mg/kg mRNA-3927 ~3, ~4, ~5, ~6, ~7, ~8, ~9, ~10, ~11, ~12, ~13   Wild type ~65, ~68, ~70, ~72, ~75, ~78, ~80, ~82, ~85, ~88     WT animals were not included as a control when assessing MUT activity in mRNA-3705 treated Mut^{-/-};Tg^{INS-CBA-G715V} hypomorphic mice. A sentence has been added to the results describing previous work (P7, Ln 142-146). While WT mice were not included in this study, subsequent studies that included unaffected heterozygous	Group	Activity (nmol/min/mg protein)	Control	~2, ~3, ~4, ~5, ~6	0.5 mg/kg mRNA-3927	~3, ~4, ~5, ~6, ~7, ~8, ~9, ~10, ~11, ~12, ~13	Wild type	~65, ~68, ~70, ~72, ~75, ~78, ~80, ~82, ~85, ~88	P7, Ln 142-146 P9, Ln 185-187 Supplemental Methods (P2-3)
Group	Activity (nmol/min/mg protein)										
Control	~2, ~3, ~4, ~5, ~6										
0.5 mg/kg mRNA-3927	~3, ~4, ~5, ~6, ~7, ~8, ~9, ~10, ~11, ~12, ~13										
Wild type	~65, ~68, ~70, ~72, ~75, ~78, ~80, ~82, ~85, ~88										

		Mut^{+/-} littermates demonstrated that treatment of Mut^{+/-}; Tg^{INS-CBA-G715V} with mRNA-3705 at 1.0 mg/kg resulted in approximately 50% of MUT activity compared with Mut^{+/-} mice 24 hours postdose (data not shown). For PKU, the Phe levels in WT mice were evaluated in a separate experiment. A sentence has been added to the results section describing this work (P9, Ln 185-187) Wild -type (WT) mice were not included in this study; however, subsequent studies including vehicle-treated WT mice showed that Phe levels were around 100 μM for the duration of the study (data not shown). A detailed description of the selected animal models and rationalizations for why they were chosen has been added to the Supplementary Methods (P2-3).	
4	Appropriate use of statistics: Results of statistical analysis and significance are not clear or lacking. The experiments shown in Fig. 3 (for hMUT) are the only ones that have indicated relative p-values. If the data in the paper are largely not significant (either between dosages or compared to controls), it would make drawing conclusions and proposing doses for FIH studies problematic. If much of the data are significant, indications of significance should be made on their respective graphs and p-values provided in the text, legends, or supplementary data. Alternatively, if AUC values are more fitting for some datasets, those values should be included, compared, and discussed. While some of the statistical methodology is discussed in methods, it is absent from most of the results.	We acknowledge the importance of statistical analyses for these studies. As proposed by the Reviewer, we feel that summary statistics, including plasma PK parameters for mRNA derived from the mRNA-therapeutics and PD parameters of the relevant biomarkers, would be most fitting. As such we have added six supplementary tables detailing the summary statistics that were collected for these studies.	Supplementary Tables 1-7
5	Conclusions: The conclusions presented are that “these data demonstrate that mRNA therapeutics delivered via LNPs can reduce biomarkers for disease in murine models, and that PK/PD models based on these findings can be used to support the selection of efficacious doses for FIH studies.” The first conclusion is valid, however, this conclusion was already shown in each of the separate publications indicated earlier. This current study validates that previous work but it also uses the same therapies, animal models, and biomarkers employed previously. In that sense, this conclusion is robust and reliable and is	We thank the Reviewer for their thorough summation of our conclusions and agree about the value of these findings in bolstering the robustness and reliability of translational research from previously published studies that promote the use of murine models to assess LNP-based therapeutics. While currently available information regarding a Phase 1/2, open-label, dose escalation study of mRNA-3927 in patients with propionic acidemia suggest that the starting doses selected in accordance with these murine models are generally well-tolerated (Moderna News Details, Sept 13, 2023) and	NA

	consistent with previous literature. The conclusion regarding selection of proper FIH dosing has proven to be valid for one of the three therapies, however, to what extent these dosing predictions are effective in clinical patients for the other two therapies remains to be seen. To that end, it is difficult to say if the second batch of conclusions are robust or reliable.	efficacious (Moderna News Details, Sept 8, 2022). A Phase 1/2 trial for mRNA-3705 is underway (clinicaltrials.gov identifier: NCT04899310); a FIH trial for mRNA-3210 has not yet begun. As such, we believe that the data presented herein, particularly with regard to the PK/PD modeling and suggested starting doses, are of value as a predictive tool to assisting decision makers as mRNA-3705 and mRNA-3210 progress to FIH trials.	
6	Suggested improvements: Two major areas of improvements would help this manuscript. Statistical analysis: a. Add p-values for experiments in legend, graphs, or in text. Indicate and discuss statistical significance, or lack thereof, of findings. Indicate significance on all relevant graphs. b. Alternatively, if AUC values are more fitting for some datasets, those values should be included, compared, and discussed. c. Perform experiments for PKU mice (Fig. 4A-C) with controls. There is a control group in Fig. 4D (blood Phe levels) but none of the others in this set of experiments. d. Compare dosing regimens. This work examines the concentration of LNP-mRNA delivered but previous preclinical studies (cited above) with these therapeutics show that repeat administration (once per week for several weeks) maintains improvements in biomarker readouts without major toxicity. For at least one or two of the dosages per therapeutic, what do the PK/PD results look like when comparing a single dose administered a week apart vs. another regimen (once every three days, for example)? This is more reflective of clinical administration in patients and better informs FIH dosing recommendations. e. Multiple times AUC (tlast) results are mentioned, but no values are provided. (p. 6 line 107, p. 7 line 126-139) Broader significance/impact: While the clinical relevance is clear, does this provide broader insights into LNP-mediated mRNA therapies, improve upon or alter our methodologies for predicting dosing, or substantially expand our understanding of these investigational therapies? To address this: f. If there is a broader impact from this work or relevance to an array of interests, clearer explanation of this should be made in the introduction and discussion.	a. As later suggested by the Reviewer, we have included summary statistics in place of reporting p values for pharmacokinetic parameters. All data exhibiting statistically significant differences are denoted on the relevant figures and graphs (e.g., P7, Ln 133-137; Figure 3b). Furthermore, language specific statistical significance is used in the main text where applicable. When non-statistically significant comparisons are made, the term “nonsignificant” is used to describe findings (e.g., P6, Ln 113). b. Seven supplementary tables detailing summary statistics have been added to the Supplementary Material. These tables are called out in the respective results section where summary statistics were initially reported in the main body text. c. mRNA analysis is specific to the mRNA test article. Therefore, no test article specific mRNA would be evaluable in vehicle treated PKU or WT mice. The assay to measure the mRNA-derived protein is specific to the human protein and therefore is not evaluable in vehicle treated PKU or WT mice. d. As part of the therapeutic development, studies with varying dose levels and dosing regimens have been conducted to evaluate improvements in biomarkers, toxicology, etc. Our approach herein incorporates the nonclinical data (mRNA, protein, biomarkers) from a single dose, while modeling captures the dose and dosing regimens for multiple doses proposed for the FIH trials. e. Summary statistics been added to Supplementary Table 1. f. Content has been added to the discussion highlighting the broader implications of these findings, such as in mitigating reliance on nonhuman primates in assessing mRNA-therapeutic efficacy and emphasizing the consistency for recommended FIH dose levels across different rare disease programs with PK/PD modeling.	a. E.g., P7, Ln 133-137; Figure 3b b. Supplementary Tables 1-7 c. NA d. NA e. Supplementary Table 1 f. P12, Ln 244-251 g. NA

	g. If there is a further refinement of study or approach using these specific therapies, that should be discussed.	g. No further refinement is planned from a modeling perspective for these specific therapies.	
7	References: In some cases, original research should be cited. For example, a review from Chandler and Venditti (Hum Gene Ther 2019) is cited as both evidence of the advantages of LNP in terms of its intracellular transport and genotoxicity and evidence of toxicity of viral gene delivery. However, this citation is a review with a sentence or two briefly stating those conclusions. Original literature or a more specific, comprehensive review should be cited.	We agree that primary manuscript should be cited when possible. In some cases, we have retained citations for review articles, since we feel this provides the most comprehensive support for statements made. For example, when defining propionic acidemia and its underlying causes in the introduction, we have cited a single review article rather than multiple primary manuscripts. Two alternative citations have been provided in place of the Chandler and Venditti (Hum Gene Ther 2019) review to support the statement that preexisting immunity to adeno-associated virus gene delivery systems can be a barrier to implementation (Harrington EA, et al. Hum Gene Ther. 2016;27(5):345-53 ; Chandler RJ, et al. J Clin Invest. 2015;125(2):870-80). Furthermore, two citations have been added describing the safety and tolerability of mRNA-LNP therapies in mouse models (An D, et al. Cell Rep. 2017;21, 3548-3558 (2017) ; Jiang L, et al. Nat Commun. 2020;11, 5339).	P5, Ln 77-82
8	Clarity and context: This study is very clear and focused. The goals and conclusions from the introduction through the results and discussion are consistent and clear. More discussion of the broader scientific or clinical significance of this work in the introduction and discussion would benefit the reader in contextualizing this work.	We agree that an enhanced discussion would be beneficial to contextualize this work for the reader. As such, we now call out numerous insights gained from these analyses, including that the reduction in biomarkers in these murine models correlated with the therapeutic benefits observed with standard-of-care treatments, the potential of murine models to mitigate reliance on nonhuman primates in assessing mRNA-therapeutic efficacy, and the relative similarity between these models for FIH starting doses.	P10-13, Ln 206-262
Reviewer #2 Comments:			
1	The manuscript describes the outcomes of treating murine models of propionic acidemia, methylmalonic acidemia, and phenylketonuria with mRNA delivered with liver-directed lipid nanoparticle (LNP) technology. Although the experimental details differ for the three individual models, the methods are sound. The manuscript is well-written. The preclinical data on the propionic acidemia model formed the rationale for a first in human clinical trial. However, the designs of the individual preclinical experiments for the three diseases are rather different. Although the common root is the investigation of mRNA therapy in inborn errors of metabolism, given that there are distinct biological and clinical differences	We thank the reviewer for their thorough summary of our model and appreciate their acknowledgement that the manuscript is well-written. Rationale for including three different models of rare disease is due in part to a request to submit this as a companion paper for a publication requested by Nature that is currently being considered for publication. That paper outlines interim findings for a first in human interim analysis of mRNA-3927 in patients with propionic acidemia. We believe this manuscript that outlines the preclinical studies used to guide FIH doses for this trial, as well as its potential to guide FIH doses for similar rare disease programs, is important to note. As such, we have added	P12, Ln 247-251

	between the disorders, the rationale for lumping all three experiments into a single manuscript is unclear.	language to the Discussion highlighting the value of reporting these models in this integrated approach, noting the consistency and relatively similar starting doses across different diseases. Furthermore, this integrated approach demonstrated consistency and relatively similar starting FIH dose levels across three different rare disease programs. As such, this approach holds promise in enhancing the accuracy of predicting FIH doses, ensuring optimal efficacy, and facilitating the rational design of clinical studies, ultimately promoting the likelihood of success in the clinical setting.	
2	To facilitate comparisons to other investigators in the field, more details regarding the design, production, and purification of the mRNA/LNP formulation should be provided.	We agree that a detailed description of the mRNA/LNP formulation is warranted to facilitate comparisons in the field. As such, a section entitled “Lipid nanoparticle production and formulation” has been added to the Supplementary Methods section characterizing of mRNA/LNP production and formulations.	Supplementary Methods, P3
3	Regarding the mRNAs, how were these synthesized? Did they incorporate modified bases and optimized untranslated regions? How were they capped? How were they purified prior to loading into LNPs?	Due to the propriety nature of the mRNA sequences utilized herein, we have not reported the sequence itself, which includes modified bases. However, a description of mRNA synthesis and production, as well as a references to previously published studies with detailed methodology for mRNA synthesis and formulation have been added to the Supplementary Methods (An D, et al. Cell Rep. 2017;21, 3548-3558 (2017); Jiang L, et al. Nat Commun. 2020;11, 5339; Sabnis S, et al. Mol Ther. 2018;26, 1509-1519).	Supplementary Methods, P3
4	Regarding the LNPs, the terminology SM86/OL-56 is used in the manuscript to identify the lipids incorporated into the LNP, but this will have no meaning to anyone who is not an aficionado of the field. The meaning of these abbreviations should be included. However, more information regarding the LNP structure is necessary. What were the final ratios of the cationic lipid: phospholipid: cholesterol: PEG? What method was used to mix the lipids with the mRNA? How were the complexes then purified? In what solution were the complexes finally suspended for intravenous injection? I understand that some of the details of the LNP structure may be proprietary, but I am not asking for sufficient information that reagents can be duplicated. However, the details I have	A section entitled “Lipid nanoparticle production and formulation” has been added to the Supplementary Methods section providing a detailed characterization of LNP formulations, including LNP particle size, encapsulation, polydispersity indices, and more.	Supplementary Methods, P3

	requested, I propose, are key to understanding both efficacy and toxicity of mRNA/LNP therapeutics. Without this information, it is impossible to compare the outcomes of different experiments in the literature.		
5	I recommend clarifying at the outset that the calculated dose given to the mouse is the amount of mRNA in mg/kg body weight delivered and does not include the mass of the LNP. Then, the molar ratio of mRNA to lipid should also be given.	A sentence has been added to the Study design and treatment groups section of the Methods clarifying that all doses were based on mRNA concentrations only, without regard for the mass of the LNPs. All doses (mg/kg) were based on mRNA concentrations only and did not consider the molecular weight of the LNP.	P14, Ln 283-284
6	Propionic acidemia The PCCA deficient models harbors a missense variant that apparently retains some PCC expression. Is it known whether the human PCCB monomer can complex with endogenous mutant mouse PCCA monomer? Or whether the human PCCA can complex with mouse PCCB? If so, this could alter the effectiveness of the therapy if one can deliver only a partial amount of message in comparison to the amount of native mRNA and PCC protein already present.	We thank the reviewer for this interesting and insightful question. To date, we have yet to assess cross-species binding between murine and human PCCA/B components. Given the homology of these subunits (murine PCCA is 89% identical to human PCCA; murine PCCB is 93% identical to human PCCB) we would hypothesize that interactions are possible. However, human PCCA should have a higher affinity for human PCCB. For this reason, we injected human PCCA and PCCB together in Pcca ^{-/-} mice to facilitate a rapid PCC complex formation, and onset of efficacy.	NA
7	The treatment did yield an increase in liver PCC activity in the PCC ^{-/-} mice over control-treated mice. How does this activity compare to that treated in wild type PCC ^{+/+} mice?	PCC activity in wild-type mice have been added to Figure 2c for comparison. 	Figure 2c
8	It is intriguing that the duration of biochemical response to therapy was 3-4 weeks in the PCC ^{-/-} model but less than one week for PKU. Do the authors have an explanation for this?	The biochemical response for mRNA therapeutics is driven largely by the half-life of the proteins for which they encode. The PCCA/PCCB complex resides in the mitochondria, which may provide protection against proteolytic cleavage, relative to	P11, Ln 226-232

		PAH, which resides in cytosol. As the duration of a biochemical response is an important consideration when developing mRNA therapeutics, this hypothesized explanation has been added to the discussion. Interestingly, the biochemical responses observed in Pcca^{-/-} (A138T) mice treated with mRNA-3927 persisted for 3 to 4 weeks, while those observed in mRNA-3210 treated PAHenu2 mice persisted for <168 hours. Since the biochemical response for an mRNA therapeutic is driven largely by the half-life of the protein(s) for which it encodes, the fact that the PCCA/PCCB complex resides in the mitochondria relative to PAH, which resides in cytosol, may provide protection from proteolytic cleavage and factor into the longevity of the therapeutic effect.	
9	Methylmalonic acidemia Again, was the liver MUT enzyme activity in treated mice compared to that of wild type mice?	We agree that information pertaining to unaffected controls is important for comparison purposes. Wild type mice were not included in this study; however, subsequent studies included unaffected heterozygous Mut^{+/-} littermates. Therein, treatment of Mut^{-/-};Tg^{INS-CBA-G715V} with mRNA-3705 at 1.0 mg/kg resulted in approximately 50% of MUT activity of Mut^{+/-} mice 24 hours post dose. This information has been added to the results section. While WT mice were not included in this study, subsequent studies that included unaffected heterozygous Mut^{+/-} littermates demonstrated that treatment of Mut^{-/-};Tg^{INS-CBA-G715V} with mRNA-3705 at 1.0 mg/kg resulted in approximately 50% of MUT activity compared with Mut^{+/-} mice 24 hours postdose (data not shown).	P7-8, Ln 142-146
10	This experiment was apparently terminated at 24 hours. Was no attempt made to study the duration of the treatment response?	This experiment was terminated at 24 hours postdose to evaluate maximum MUT protein expression and activity, as well as maximal reductions in plasma methylmalonic acid levels, based on previous experience (An et al. 2017, An et al	NA

		2019). Subsequent experiments evaluated the duration of plasma methylmalonic acid reductions following treatment with mRNA-3705. As this data was not reported herein, no additions have been made to the text.	
11	Phenyketonuria I note that hPAH mRNA was detectable in blood for much longer in this experiment than was the case for mRNA in the propionic acidemia experiment. Is there a true biological difference here or this a measurement artefact due to the differences in the sensitivity in the methods used to detect mRNA in the two experiments?	The difference in mRNA detection between these 2 programs is likely due to the difference in the sensitivity of the methods utilized to detect mRNA as branched DNA was utilized to detect hPCCA and hPCCB while hPAH was detected via RT-PCR.	NA
12	24 hours post dose, the mean concentration of hPAH mRNA measured in liver was 51.57 ng/g. First, one assumes this is ng/g liver wet weight but I could not find the definition. If the units are ng/g liver tissue, then I calculate that approximately 0.5% of the delivered mRNA dose remains in the liver at 24 hours. Is the fate of the other 99.5% of the delivered dose known?	The main text describing PK parameter summary statistics for serum and liver mRNA from mRNA-3210 have been revised to include parameters observed in both male and female mice (Supplementary Tables 5 and Supplementary Table 6 , respectively). Units for all values are provided in the tables. Furthermore, the Methods has been revised to clarify that liver weight represented the wet weight of the tissue. Liver samples from 2 mice (15-25 mg each) were collected, rinsed in 1X PBS, dried with sterile drape, weighed (wet weight), and placed into RNAase-free tubes for mRNA analysis. mRNA has a rapid half-life, which can range from minutes to hours based on a variety of complex factors. We acknowledge that a single timepoint to evaluate tissue concentrations of mRNA may underrepresent the total mRNA delivered to that tissue; however, we feel that a discussion of this is beyond the scope of this manuscript.	P15, Ln 315-316
13	How does this amount of hPAH mRNA compare to native mPAH mRNA expression?	mRNA analysis was specific to the mRNA test article (ie., hPAH mRNA). We did not evaluate native mPAH mRNA.	NA
14	After mRNA delivery, hPAH protein production clearly increased and blood phenylalanine concentrations decreased demonstrating that the protein had PAH activity, but were any attempts made to actually measure enzymatic PAH activity in liver homogenates from the treated PKU mice?	PAH activity was not directly evaluated in the liver homogenates from the mRNA-3210 treated PAH ^{enu2} mice. Rather, protein expression and biomarker reduction were evaluated as an indirect marker of activity. It is well-established that there is pronounced and sustained elevated Phe in	NA

		circulation in the PAH ^{enu2} mice. As such, activity was inferred through changes in protein expression and biomarker reductions.	
15	PK/PD modeling Why do the no-observed-effect-limits differ between the different disease models? How were these determined?	The no-observed-adverse-effect-limit (NOAEL) differed slightly between the programs due to differences in study designs, including the dose levels selected for evaluation in their respective safety studies and the sequence of when the studies were conducted. In all studies, the NOAEL was the mid-dose level in the safety studies. This clarification has been added to the “Approach to the FIH projections for mRNA-3927, mRNA-3705, and mRNA-3210” section of the Methods. No -observed-adverse-effect-limits represent the mid-dose levels observed in the safety studies. Differences between the NOAEL in the studies do not indicate the safety profiles were different. Rather, it is more reflective of the dose levels evaluated in the individual studies.	P19, Ln 416-417
16	Conclusions The ultimate goal of all three clinical projects was ultimately to determine an ideal path forward to clinical trial for each mRNA therapeutic. The PK/PD modeling providing idealized dose estimates for each mRNA, but the manuscript does not mention the predicted necessary dosing frequency. Perhaps this can ultimately only be optimized in humans, but certainly these experiments must suggest a likely initial necessary dosing frequency.	We acknowledge the importance of this information and thank the reviewer for the suggestion. Recommended FIH dosing frequencies for mRNA-3927, mRNA-3705, and mRNA-3210 have been added to the discussion. Allometric scaling of PK parameters for mRNA-3927, mRNA-3705, and mRNA-3210 indicated that 0.3 mg/kg every 3-weeks, 0.1 mg/kg every 3-week, and 0.4 mg/kg/week, respectively, were associated with initial start of plateauing in 2-MC, methylmalonic acid, and Phe levels.	P12, Ln 242-244

REVIEWERS' COMMENTS

Reviewer #2 (Remarks to the Author):

The authors have adequately satisfied all of my prior critiques.

Reviewer #3 (Remarks to the Author):

All comments are addressed. I recommend publication